# EP4-induced mitochondrial localization and cell migration mediated by CALML6 in human oral squamous cell carcinoma
Soichiro Ishikawa[1,2,3], Masanari Umemura [1,3] ✉, Rina Nakakaji[1,2], Akane Nagasako[1], Kagemichi Nagao[1], Yuto Mizuno[1], Kei Sugiura[2], Mitomu Kioi[2], Kenji Mitsudo[2] & Yoshihiro Ishikawa[1]

Lymph node metastasis, primarily caused by the migration of oral squamous cell carcinoma (OSCC) cells, stands as a crucial prognostic marker. We have previously demonstrated that EP4, a subtype of the prostaglandin E2 (PGE2) receptor, orchestrates OSCC cell migration via $Ca^{2+}$ signaling. The exact mechanisms by which EP4 influences cell migration through $Ca^{2+}$ signaling, however, is unclear. Our study aims to clarify how EP4 controls OSCC cell migration through this pathway. We find that activating EP4 with an agonist (ONO-AE1-473) increased intracellular $Ca^{2+}$ levels and the migration of human oral cancer cells (HSC-3), but not human gingival fibroblasts (HGnF). Further RNA sequencing linked EP4 to calmodulin-like protein 6 (CALML6), whose role remains undefined in OSCC. Through protein-protein interaction network analysis, a strong connection is identified between CALML6 and calcium/calmodulin-dependent protein kinase kinase 2 (CaMKK2), with EP4 activation also boosting mitochondrial function. Overexpressing EP4 in HSC-3 cells increases experimental lung metastasis in mice, whereas inhibiting CaMKK2 with STO-609 markedly lowers these metastases. This positions CaMKK2 as a potential new target for treating OSCC metastasis. Our findings highlight CALML6 as a pivotal regulator in EP4-driven mitochondrial respiration, affecting cell migration and metastasis via the CaMKK2 pathway.

Oral cancer is one of the most common malignant tumors of the head and neck[1]. Approximately 90% of all oral malignancies are squamous cell carcinomas (OSCC)[2]. Despite treatment advancements, the prognosis remains dismal, with a 5-year survival rate of about 50% in the USA. The presence of cervical lymph node metastasis is a critical prognostic factor[1,2]. Current treatments, including surgery, often lead to remarkable oral dysfunction, highlighting the need for more effective, less invasive therapies.

EP4 prostanoid receptors, a subtype of prostaglandin E2 receptors, play a pivotal role in regulating cell migration and metastasis in OSCC, among other cancers[3]. High EP4 expression correlates with increased cell migration and metastasis, mainly by modulating intracellular calcium ($Ca^{2+)}$ levels[4]. Our research indicates that EP4 stimulation enhances $Ca^{2+}$ influx via Orai1, crucial for store-operated $Ca^{2+}$ entry (SOCE), a key mechanism for extracellular $Ca^{2+}$ intake, particularly overexpressed in invasive OSCC cells[5,6]. This mechanism's significance is further highlighted by its implications in

the progression of metastatic melanoma[7,8], underscoring SOCE's role in cancer biology.

Furthermore, we have identified a novel pathway of $Ca^{2+}$ regulation in OSCC, where EP4 activates phosphatidylinositol 3'-kinase (PI3K), triggering $Ca^{2+}$ influx through Orai1 independently of traditional calcium sensors like stromal interaction molecules (STIMs)[9]. This pathway differs from those initiated by G protein-coupled receptors and immunoreceptors, which typically involve IP3-induced $Ca^{2+}$ release from the endoplasmic reticulum.

Calcium serves as a secondary messenger in various cellular processes, including signal transduction, apoptosis, and proliferation[10]. Calmodulin (CaM), a primary intracellular receptor for $Ca^{2+}$, interacts with $Ca^{2+}$/CaM-dependent kinases (CaMKs), further activated by $Ca^{2+}$/CaM-dependent kinase kinase (CaMKK1 and CaMKK2)[11]. We found that EP4 activation of CaMKK2 in OSCC cells imparts a novel influence on AMP-

[1]Cardiovascular Research Institute, Yokohama City University Graduate School of Medicine, Yokohama, Kanagawa, Japan. [2]Department of Oral and Maxillofacial Surgery, Yokohama City University Graduate School of Medicine, Yokohama, Japan. [3]These authors contributed equally: Soichiro Ishikawa, Masanari Umemura. ✉e-mail: umemurma@yokohama-cu.ac.jp

activated protein kinase (AMPK) signaling, a key regulator of cellular energy homeostasis[12–15].

In addition, our focus includes CaM-like proteins (CMLs), particularly CALML6, initially identified from Bothrops insularis[16]. The Human Protein Atlas documents that humans express four CML isoforms: CALML3, CALML4, CALML5, and CALML6. Despite this, the exact functions of CMLs remain largely unexplored. In the current study, our findings suggest that CALML6, characterized by its $Ca^{2+}$-binding capacity, emerges as a critical modulator of OSCC cell behavior.

Additionally, mitochondria, crucial energy producers in cells, have been implicated in cancer progression and metastasis[17]. Mitochondrial oxidative phosphorylation (OXPHOS) generates reactive oxygen species (ROS). ROS play important roles in cancer cell signaling and proliferation and in the regulation of apoptosis, thus affecting processes such as cell invasion and migration. The production of ROS leads to the promotion of cell invasion and metastasis in breast cancer cells and lung cancer cells[18,19]. Mitochondria are generated via mitochondrial biogenesis and eliminated by mitophagy[20]. Peroxisome proliferator-activated receptor γ coactivator-1 α (PGC-1α) regulates mitochondrial biogenesis[21]. Furthermore, in cancer cells, PGC-1α is regulated to promote OXPHOS and mitochondrial biogenesis and increase the oxygen consumption rate (OCR), resulting in the production of ROS.

The relationship between EP4 and downstream $Ca^{2+}$ signaling in cancers, particularly OSCC, has not thoroughly explored. This study aims to elucidate the link between EP4-induced $Ca^{2+}$ influx and downstream signaling in OSCC cells. We discovered that CALML6 modulates EP4-induced cell migration. Furthermore, we observed that the overexpression of EP4 in HSC-3 cells markedly enhanced experimental lung metastasis in mice. Conversely, administration of STO-609, a CaMKK2 inhibitor, greatly attenuated these metastases.

Through these findings, we highlight the EP4/CALML6/CaMKK2/AMPK pathway as a novel contributor to OSCC progression, suggesting new avenues for therapeutic intervention aimed at mitigating cell migration and metastasis.

## Results

### The mRNA transcript and protein expression levels of EP4 in human OSCC cells were higher than those in normal oral cells

In our earlier report, we reported that EP4 was expressed in human OSCC cell lines (HSC-3 and OSC-19)[9]. However, the differences in EP4 protein expression between normal cells and OSCC cells remain elusive. Therefore, we first investigated the protein expression of EP4 in human gingival fibroblasts (HGnF) and a human OSCC cell line (HSC-3). EP4 protein expression in HSC-3 cells was higher than that in HGnF cells (Fig. 1A). Furthermore, we compared the protein expression of EP4 in human oral keratinocytes (HOK), HGnF, HSC-3, OSC-19, and human glioblastoma cells (LN229) (Supplementary Fig. 1A, B). The expression of EP4 in the cancer cell lines was higher than that in the normal cell lines. Additionally, upon comparison among the primary tumor cell line SAT cells, low metastatic potential HSC-4 cells, and high metastatic potential HSC-3 cells, it was found that EP4 expression increased with the increases in metastatic capability (Supplementary Fig. 1C)[22,23].

Intracellular $Ca^{2+}$, a ubiquitous second messenger, regulates many cellular processes, including cell migration[10]. We previously confirmed that ONO-AE1-437, an EP4 agonist, notably increased the intracellular $Ca^{2+}$ concentration in HSC-3 cells, resulting in the promotion of cell migration. Therefore, we compared the increase in intracellular $Ca^{2+}$ in HGnF and HSC-3 cells upon stimulation with an EP4 agonist. The EP4 agonist rapidly increased the intracellular $Ca^{2+}$ concentration in HSC-3 cells but not in HGnF cells (Fig. 1B and Supplementary Fig. 2A). Fluo 4-AM imaging of intracellular $Ca^{2+}$ showed the same trend (Supplementary Fig. 2B). Furthermore, EP4 agonist treatment promoted the migration of HSC-3 cells but not HGnF cells, according to the scratch assay (Fig. 1C and Supplementary Movie 1). The EP4 agonist increased

both the cell migration speed and the migration distance in HSC-3 cells (Supplementary Fig. 3A, B, C). On the other hand, the EP4 agonist did not affect cell proliferation in either of the two cell lines (Supplementary Fig. 3D, E). Interestingly, similar results were also observed in HOK cells (Supplementary Fig. 4). These results suggested that EP4 regulated the intracellular $Ca^{2+}$ concentration and the migration of OSCC cells but not normal human oral cells. We summarized the mechanism in a diagram (Fig. 1D).

### Overexpression of EP4 promoted the migration of oral cancer cells

We previously reported that EP4 regulated the migration of OSCC cells through an increase in intracellular $Ca^{2+}$ and the phosphatidylinositol-3 kinase (PI3K) pathway[9]. To further confirm the relationship between EP4 and cell migration, we established OSCC cell lines overexpressing EP4 using a lentivirus system (Supplementary Fig. 5A, B, C). The scratch assay showed that the overexpression of EP4 promoted the migration of HSC-3 cells for 10 h (Fig. 2A and Supplementary Movie 2). To confirm the role of EP4 in cell invasion, we performed a Boyden chamber assay (invasion assay). This result showed that EP4 overexpression increased the number of invading cells. These results suggested that EP4 promoted both cell migration and cell invasion (Fig. 2B). In contrast, EP4 overexpression did not change cell proliferation (Supplementary Fig. 5D). Furthermore, we evaluated the number of lamellipodia in HSC-3 cells by immunocytochemistry. EP4 overexpression increased the number of lamellipodia, which reflected the activity of actin assembly and disassembly (Fig. 2C).

Next, we confirmed the cell migration ability in vivo. We established an orthotopic xenograft tongue tumor model in a prior study[24]. In this study, we implanted the EP4-overexpressing HSC-3 cells expressing green fluorescent protein (GFP) and luciferase into the tongues of mice, and determined the number of lymph node metastases with a fluorescence microscope and an in vivo bioluminescence imaging system (IVIS). Surprisingly, we observed a robust increase in the lymph node metastasis rate in the EP4-overexpressing HSC-3 cell line compared to the control HSC-3 cell line, indicating that overexpression of EP4 enhances the ability for lymph node metastasis (Fig. 2D–F). These results showed that EP4 was also involved in lymph node metastasis in vivo.

### CALML6 is related to EP4-induced migration of oral cancer cells

To identify the EP4-related molecules, we performed RNA-seq analysis in HSC-3 cells treated with/without the EP4 agonist (Supplementary Fig. 6). The results showed that the EP4 agonist greatly increased the mRNA transcription of calcium-binding proteins 3 h after stimulation. Among the 62 upregulated genes, the mRNA transcription of CaM-like protein (CMLs) 6 (CALML6, also known as CAGLP) was increased by EP4 agonist stimulation (Fig. 3A). CMLs are different forms of the CaM proteins among the members of the EF-hand protein family[16]. Chen et al. cloned CALML6 and characterized it in 2004[25]. The function of CALML6 is still unknown. Interestingly, CALML6 is highly expressed in skeletal muscle and tongue, according to the Human Protein Atlas. In the current study, the mRNA transcript level of CALML6 in HSC-3 cells was higher than that in HGnF cells (Fig. 3B). The CALML6 protein expression level in HSC-3 cells was also higher than that in HGnF cells (Fig. 3C). Furthermore, compared to the normal cells HOK, the expression of CALML6 was higher in oral cancer cells, particularly in the high metastatic potential HSC-3 cell line, where it was most pronounced (Supplementary Fig. 7). The protein expression level of CALML6 in tongue tumor tissue was higher than that in adjacent normal tongue tissue, as determined by immunohistochemistry (Fig. 3D and Supplementary Fig. 8). Furthermore, knockdown of CALML6 by shRNA using lentiviral transduction decreased the migration of HSC-3 cells, suggesting that CALML6 is involved in the EP4-induced migration of OSCC cells (Fig. 3E, Supplementary Fig. 9A, B and Supplementary Movie 3). In contrast, knockdown of CALML6 did not change cell proliferation (Supplementary Fig. 9C).

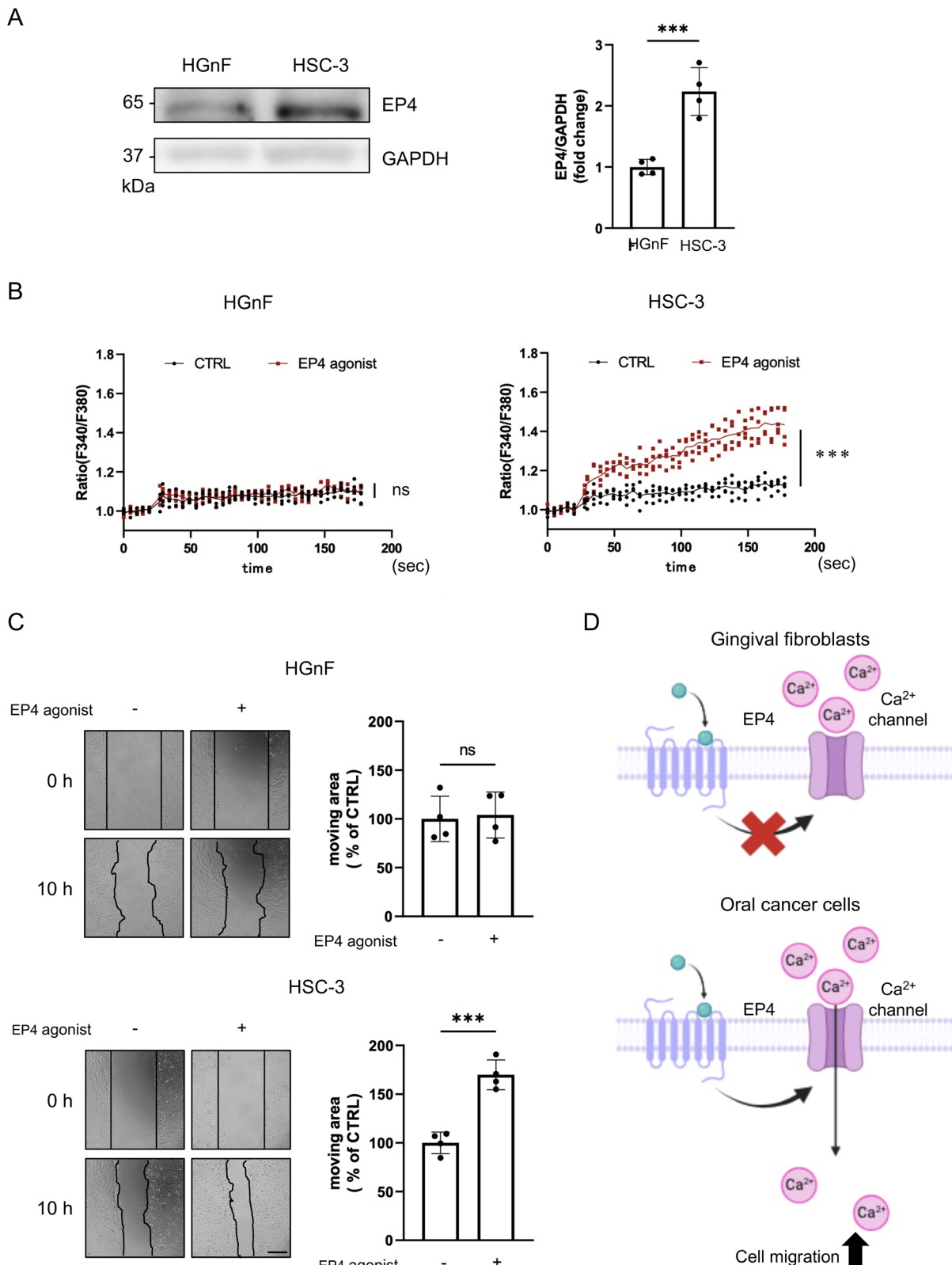

**Fig. 1 | EP4 induced an increase in Ca²⁺ and promoted cell migration in OSCC cell lines but not in normal cells. A** Protein expression of EP4 in human gingival fibroblasts cells (HGnF cells) and human OSCC cells (HSC-3). **B** The EP4 agonist induced a significant increase intracellular Ca²⁺ in HSC-3 cells but not HGnF cells. **C** Representative pictures and quantitative analysis of the scratch assay. These results confirmed that the EP4 agonist promoted the migration of HSC-3 cells but not HGnF cells. **D** Proposed mechanism of the EP4 signaling-mediated increases in the Ca²⁺ concentration and cell migration in HGnF cells and OSCC cells. The schema was created using BioRender. **A**–**C** Data are representative of $n = 4$ independent experiments. **A**–**C** Unpaired $t$ test; ns, not significant; **$p < 0.01$, ***$p < 0.001$. Source data and exact $p$ values are provided as a source data file.

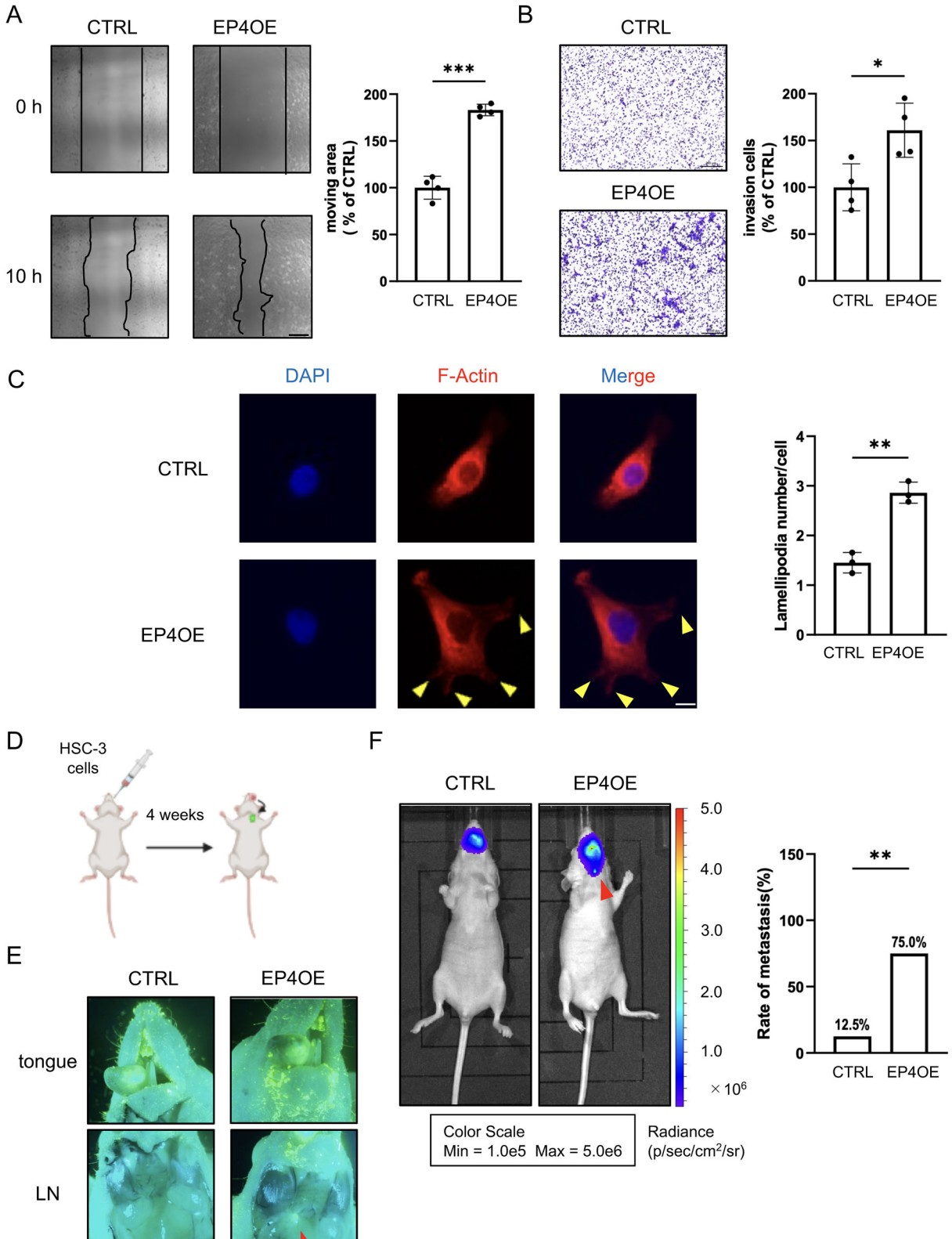

## EP4 activated AMPK via phosphorylation of CaMKK2

As mentioned above, our results showed that EP4 might regulate cell migration via CALML6. This hypothesis is reasonable, because we already confirmed that EP4 agonist treatment increased the intracellular $Ca^{2+}$ concentration, and there are many reports that $Ca^{2+}$ signaling regulates diverse cellular functions, including cancer cell migration[10].

First, we constructed a protein–protein interaction (PPI) network using the STRING database to investigate CALML6-related proteins[26]. This result indicated that CALML6 has a direct association with CaMKK2 (Fig. 4A). Therefore, we next focused on CaMKK2 to investigate the mechanism of EP4-induced cell migration. We confirmed that treatment with the EP4 agonist (1 μM) resulted in phosphorylation of CaMKK2 (Ser511) (Fig. 4B).

**Fig. 2 | EP4 was involved in the migration and invasion of OSCC cells in vitro and in vivo. A** Representative pictures and quantitative analysis of the scratch assay. These results confirmed that the overexpression of EP4 promoted the migration of HSC-3 cells. **B** The Boyden chamber assay (invasion assay) showed that overexpression of EP4 also promoted the invasion of HSC-3 cells. **C**, Overexpression of EP4 increased the number of lamellipodia in HSC-3 cells. The left panel shows DAPI staining of HSC-3 cells. The middle panels show F-actin staining. The yellow arrows indicate lamellipodia. The right panel shows images of double staining with DAPI (*blue*) and immunostaining for F-actin (*red*). The graph shows the number of lamellipodia in HSC-3 cells. More than 50 cells were counted per field of view. **D** Illustrating the schema for the animal experiment. We inoculated HSC-3 cells into the left lateral margin of the tongue in mice, dividing them into the CTRL group and the overexpression of EP4 group to observe the lymph node metastasis. The schema was created using BioRender. **E** Representative images of lymph node metastasis represented by GFP fluorescence detected by fluorescence microscopy. The red arrows indicate lymph node metastasis. EP4 was involved in lymph node metastasis in vivo. Overexpression of EP4 increased the rate of lymph node metastasis in mice. **F**, Representative images of lymph node metastasis represented by luciferase expression detected by IVIS. The red arrows indicate lymph node metastasis. Data are representative of $n = 3$ (**C**), $n = 4$ (**A, B**), and $n = 8$ (**E, F**) independent experiments. **A–C, E, F** Unpaired $t$ test; *$p < 0.05$, **$p < 0.01$, ***$p < 0.001$. Source data and exact $p$ values are provided as a source data file.

We also confirmed that treatment with the EP4 agonist resulted in CaMKK2 phosphorylation 15 min after stimulation (Fig. 4C). It is well known that AMP-activated protein kinase (AMPK) signaling is involved in metabolic regulation and is activated by the upstream kinase CaMKK via phosphorylation at threonine 172[15]. We also confirmed that the EP4 agonist resulted in AMPK (Thr172) phosphorylation 15 and 30 min after stimulation (Fig. 4D). Furthermore, stimulation with EP4 also increased the phosphorylation of myosin light chain (MLC) (Supplementary Fig. 10). Furthermore, knockdown of Orai1 and treatment with YM-58483 (a SOCE inhibitor) inhibited the EP4 agonist-induced phosphorylation of CaMKK2 and AMPK (Supplementary Fig. 11). Knockdown of CALML6 by shRNA using lentiviral transduction also inhibited the EP4 agonist-induced phosphorylation of CaMKK2 and AMPK (Supplementary Fig. 12).

## CaMKK2 promoted the migration of OSCC cells through phosphorylation of AMPK

As mentioned above, CaMKK2 may play a major role in EP4-mediated cell migration via phosphorylation of AMPK. Therefore, we examined the changes in downstream signaling and the cell migration ability upon CaMKK2 inhibition. As expected, STO-609, a CaMKK2 inhibitor, prevented the EP4 agonist-induced phosphorylation of AMPK and prevented cell migration in the scratch assay (Fig. 5A, B and Supplementary Movie 4). Furthermore, we established two CaMKK2 knockdown HSC-3 cell lines with a shRNA lentivirus system. Knockdown of CaMKK2 also inhibited both EP4-induced AMPK phosphorylation and cell migration (Fig. 5C, D, Supplementary Fig. 13 and Supplementary Movie 5). In contrast, knockdown of CaMKK2 did not change cell proliferation (Supplementary Fig. 13). Taken together, these findings indicated that EP4 promoted cell migration via CALML6/CaMKK2/AMPK signaling in OSCC cells.

In our earlier study, we demonstrated that inhibiting EP4 reduced experimental lung metastasis in mice with OSCC cells[9]. Following our protocol, we injected the EP4-overexpressing HSC-3 cells into the tail veins of mice and intraperitoneally administered STO-609 (30 μg/kg body weight) twice per week[27]. After 3 weeks, we evaluated experimental lung metastasis by IVIS imaging[28]. Overexpression of EP4 increased experimental lung metastasis in mice. In contrast, treatment with STO-609 inhibited experimental lung metastasis, suggesting that CaMKK2 might be a novel target for the treatment of OSCC metastasis (Fig. 5E–G). We confirmed the same trend in GFP expression by fluorescence microscopy (Fig. 5H)[29]. Furthermore, overexpression of EP4 increased the lung weight, an increase that was prevented by STO-609 (Fig. 5I).

## EP4 promoted mitochondrial biogenesis and increased the mitochondrial respiratory capacity

It is reported that AMPK regulates metabolic and energy homeostasis in cancer cells. AMPK regulates the expression of PGC1-α, which controls mitochondrial biogenesis and promotes OXPHOS, resulting in the production of ATP in cancer cells[30]. Mitochondria are involved in cell migration, invasion, and metastasis[17,31]. To evaluate mitochondrial biogenesis, we evaluated mitochondria-related genes. PGC1-α is a transcriptional coactivator that is a key inducer of mitochondrial biogenesis in cells. PGC1-α influences mitochondrial respiration, the ROS defense system, and fatty acid metabolism by interacting with specific transcription factors[32]. PGC1-α plays an important role as a stress sensor in cancer cells and can be activated by nutrient deprivation, oxidative damage, and exposure to chemotherapeutic agents. In the current study, the EP4 agonist increased the mRNA transcription and protein expression of PGC1-α (Fig. 6A). The EP4 agonist also increased the mitochondrial DNA (mtDNA) content and mitochondrial transcription factor A (TFAM) expression (Fig. 6B). Furthermore, immunocytochemistry showed that the EP4 agonist resulted in localization of mitochondria to the edge of cells (Fig. 6C). Overall, our results demonstrated that EP4 induced mitochondrial localization and cell migration mediated by CALML6 in human OSCC cells.

We sought to determine the mitochondrial function by which EP4 signaling promotes the migration of HSC-3 cells. We measured parameters related to energy metabolism in HSC-3 cells with a XF Cell Mito Stress Test Kit. This kit can be used to measure glycolytic activity by determining the extracellular acidification rate (ECAR) and measure mitochondrial OXPHOS activity on the basis of the oxygen consumption rate (OCR) through real-time and live-cell analysis[33]. Among the mitochondrial respiration-related parameters, the spare respiratory capacity (SRC) is a particularly robust functional parameter for evaluating mitochondrial reserve[34]. The relative value of the SRC was calculated by the following formula: Maximal OCR/(Basal OCR *100). EP4 agonist treatment increased the SRC and maximal OCR in mitochondria (Fig. 6D). EP4 agonist treatment for 3 h also increased ATP production, according to the ATP assay results (Fig. 6E). EP4 agonist treatment for 3 h increased ROS production, as determined by both measurement of DCFH-DA fluorescence and electron spin resonance (ESR) spectroscopy, which is recognized as one of the most powerful methods to detect ROS (Fig. 6F, G). Knockdown of CALML6 suppressed the EP4-indeuced ROS production in OSCC cells (Supplementary Fig.14). Multiple endogenous and exogenous factors promote ROS production, which leads to various physiological effects[35]. At low levels, ROS act as intracellular second messengers. In contrast, moderate levels of ROS are beneficial because they increase metabolism and growth signaling in cancer cells. Furthermore, to further investigate the relationship between EP4 agonist-induced ROS production and cell migration, we evaluated the effects using N-Acetyl-L-cysteine (NAC), an antioxidant. As expected, the enhancement of cell migration induced by EP4 agonist stimulation was inhibited by NAC (Supplementary Fig. 15 and Supplementary Movie 6). Based on our results, EP4 promoted mitochondrial respiration through mitochondrial biogenesis, resulting in ROS production. In turn, ROS promoted the migration of OSCC cells.

## Discussion

Our discussion revisits the discovery of CALML6, a putative $Ca^{2+}$-binding protein with EF-hand motifs, first identified in the venom glands of Bothrops insularis by Junqueira-de-Azevedo et al. in 2003, and later characterized in human skeletal muscle by Chen et al. in 2004[25,36]. Our study adds to this body of knowledge by not only confirming the mRNA transcription and protein expression of CALML6 in OSCC cells and normal oral cells but also by uncovering its novel role in regulating cancer cell migration. This finding is significant given the previous reports' emphasis on CALML6's role in skeletogenesis, carbohydrate/glucose metabolism, the olfactory pathway, and antiviral innate immunity, with limited exploration in mammals.

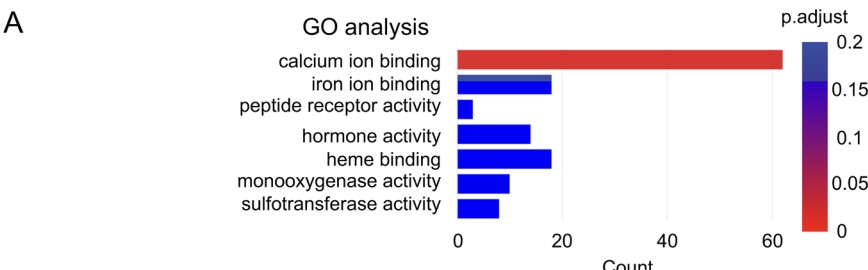

**A** GO analysis

calcium ion binding (62 genes)

SCIN, PRSS3, FSTL4, CDH17, PCDHGA2, DLL3, CRTAC1, TNNC2, ANXA13, THBS4, EFCC1, KCNIP3, NID1, MAN1C1, COLEC11,CASQ2, PCDHB12, MATN4, MASP1, CIB2, OIT3, GUCA1C, FBLN5, RCN3, CDH18,VWDE, DMP1, MICU3, MYL3, ADGRL4, HPGDS, IHH, SPTA1, S100P, FAT3, CDH16,CALML6, RASGRP4, EFCAB12, ADGRE1,SCUBE2, ITLN1, F2, PROS1, EFCAB6,PLCD1, SLIT1, PLA2G2C, RYR1, HRNR, EFCAB9, EFCAB8, BGLAP, PCDHGC4, ACTN3, PCDHA10, PCDHGB6, PCDHGA5, PCDHGA6, PCDHGB4, PCDHGA3, PCDHGB3

**Fig. 3 | CALML6 was expressed in OSCC cells and OSCC human tissues and promoted cell migration. A** RNA-seq analysis of HSC-3 cells treated with/without the EP4 agonist. GO analysis showed that EP4 agonist treatment upregulated 62 genes related to calcium ion binding, including CALML6. **B** mRNA transcription of CALML6 in HGnF cells and HSC-3 cells. **C** Protein expression of CALML6 in HGnF cells and HSC-3cells. **D** Immunohistochemical staining of CALML6 in an OSCC tissue microarray. Protein expression of CALML6 in adjacent normal tongue tissue and human tongue tumor tissue. **E** Representative images and quantitative analysis of the scratch assay. These results confirmed that knockdown of CALML6 suppressed the migration of HSC-3 cells. **A**–**C**, **E** Data are representative of $n = 4$ independent experiments. **B**, **C** Unpaired $t$ test, **E** One-way ANOVA, Tukey's multiple comparisons test; $*p < 0.05$, $**p < 0.01$, $***p < 0.001$. Source data and exact $p$ values are provided as a source data file.

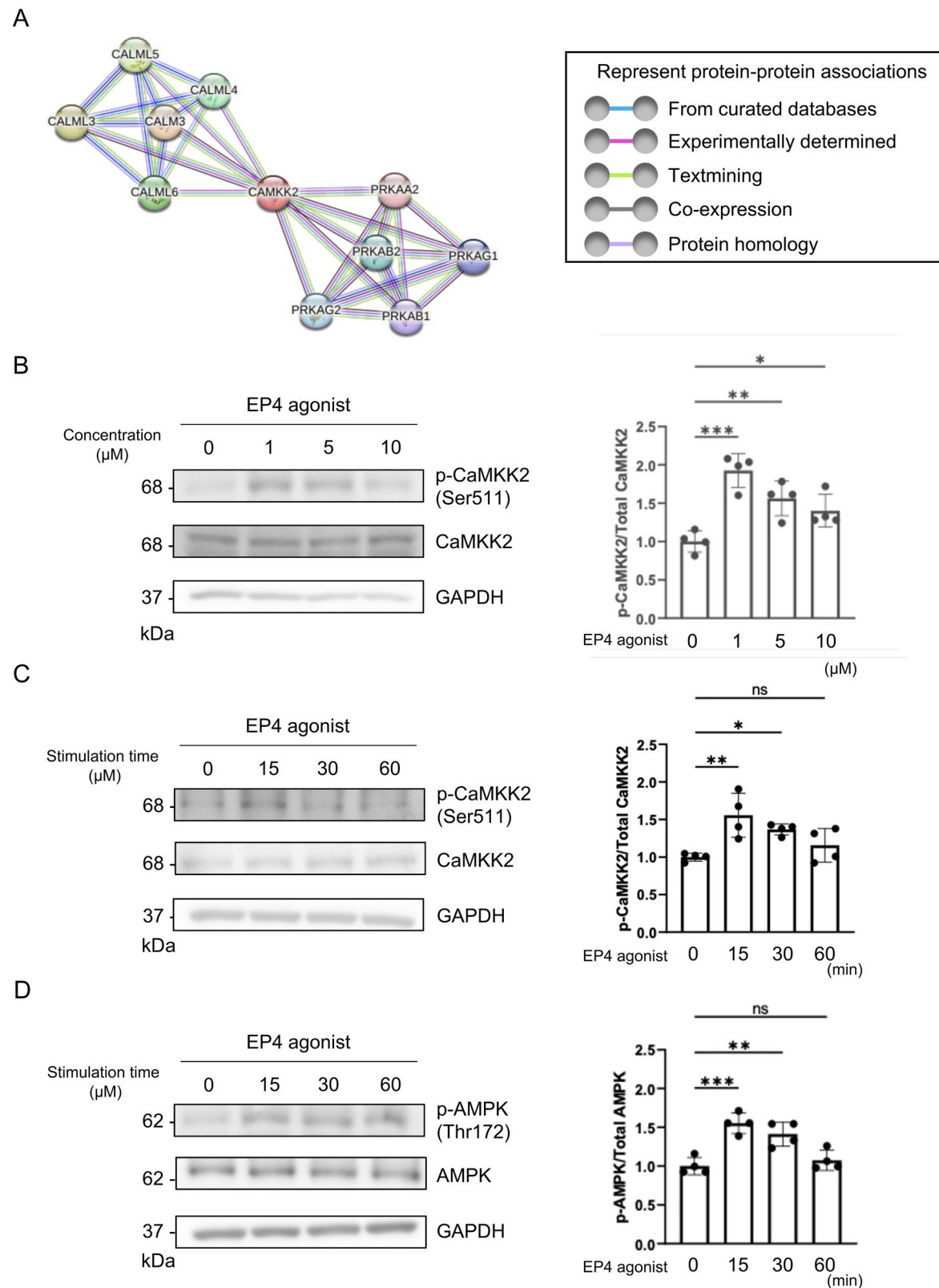

In the current study, EP4 agonist treatment resulted in phosphorylation of CaMKK2 in OSCC cells. Najor et al. demonstrated that CaMKK2 is highly expressed in gastric cancer and leads to its progression[14]. The authors reported that STO-609 decreased the migration and invasion of gastric cancer cells (AGS cells). These data are consistent with the findings in our OSCC cells. In the current study, overexpression of EP4 promoted the migration of OSCC cells. Knockdown of EP4, CALML6 or CaMKK2 suppressed the migration but not the proliferation of OSCC cells. Lin et al. reported that the expression of CaMKK2 was upregulated in hepatocellular carcinoma (HCC) and negatively correlated with HCC patient survival[28]. They showed that knockdown of CaMKK2 attenuated liver cancer cell growth in vitro and in vivo. Furthermore, they

**Fig. 4 | CALML6 was associated with CaMKK2 in OSCC cells. A** Protein–protein interaction (PPI) network analysis showed that CALML6 was associated with the CaMKK family, whose members are serine/threonine protein kinases, and with the $Ca^{2+}$/calmodulin-dependent protein kinase subfamily and phosphoribulokinase (PRK) family, whose members are enzymes unique to the reductive pentose phosphate pathway of $CO_2$ assimilation. **B** Representative images of CaMKK2 in the presence of the EP4 agonist are shown (*left*). Densitometric analysis of western blot bands showed that EP4 agonist treatment (1 µM) resulted in the greatest increase in CaMKK2 (Ser511) phosphorylation (*right*). **C** Representative images of CaMKK2

(Ser511) in the presence of the EP4 agonist are shown (*left*). Densitometric analysis of western blot bands showed that EP4 agonist treatment (1 µM) increased CaMKK2 phosphorylation 15 min after stimulation (*right*). **D** Representative images of AMPK (Thr172) in the presence of the EP4 agonist are shown (*left*). Densitometric analysis of western blot bands showed that EP4 agonist treatment (1 µM) increased AMPK phosphorylation 15 and 30 min after stimulation (*right*). **B–E** Data are representative of $n = 4$ independent experiments. **B–E** One-way ANOVA, Tukey's multiple comparisons test; ns, not significant; *$p < 0.05$, **$p < 0.01$, ***$p < 0.001$. Source data and exact $p$ values are provided as a source data file.

demonstrated that administration of STO-609 (30 µg/kg body weight) twice per week for 4 weeks decreased the tumor volume by approximately 21%, while the volume of tumors in vehicle-treated mice increased by approximately 50%. In our study, the same concentration of STO-609 decreased the lung metastasis of HSC-3 cells in vivo.

Long YC et al. demonstrated that AMPK signaling is involved in metabolic regulation and is activated by the upstream kinase CaMKK via phosphorylation of threonine 172[15]. AMPK regulates the expression of PGC1-α, which controls mitochondrial biogenesis by activating p38 and inhibiting mTOR in cancer cells to regulate oxidative metabolism and maintain the ATP pool[30,37]. PGC1-α regulates the expression of nuclear genes for respiratory chain function, transcription, and replication of mtDNA by activating transcription factors (NRF-1 and NRF-2), tumor suppressor genes (SIRT3), nuclear-encoded mitochondrial enzymes (POLRMT), and TFAM. The metastatic and migratory capabilities of ovarian cancer cells are also reported to be influenced by the distribution of mitochondria at the leading edge, regulated through AMPK signaling[38]. In our current study, treatment with an EP4 agonist led to the phosphorylation of CaMKK2 and AMPK. This, in turn, activated PGC1-α and TFAM, enhancing ATP production and its distribution at the leading edge.

In their examination of EP4 and mitochondrial interactions, Ying et al. investigated EP4's role in cardiac fatty acid metabolism, finding that EP4 deficiency leads to heart dysfunction marked by hypertrophy and fibrosis due to impaired fatty acid uptake and ATP generation[39]. Furthermore, they showed that EP4 deletion boosts metabolic rate and mitochondrial activity in white adipose tissue, leading to decreased fat mass and modified lipid storage via the AMPK-fat-specific protein (FSP)-27 pathway[40]. While these studies highlight EP4's function in normal tissues, there remains a notable gap in the literature regarding EP4's role in mitochondrial function within cancer cells. Mitochondrial metabolism adapts to stress conditions and maintains bioenergetic processes related to cellular functions[34]. In our study, EP4 agonist treatment increased the mitochondrial spare respiratory capacity (SRC). The SRC depends on the integrity of the mitochondrial electron transport chain and the permeability of the inner mitochondrial membrane to protons. Overall, we hypothesized that EP4 promoted the consumption of a large amount of oxygen for OXPHOS in mitochondria, resulting in excessive ROS production in cancer cells. Tochhawang et al. reported that ROS activates the Mitogen-activated Protein Kinase (MAPK) signaling cascade and nuclear factor kappa-light-chain-enhancer of activated B cells (NF-κB) pathway, leading to the activation of target genes and epithelial-mesenchymal transition (EMT) genes, which results in cell migration, invasion, and EMT[41]. Indeed, we confirmed that EP4 agonist treatment promoted ROS production, based on both fluorescence microscopy and ESR spectroscopy. Chae et al. demonstrated that the EP4-specific agonist CAY10598 reduced cell viability and induced apoptosis to promote ROS production through signal transducer and activator of transcription 3 (STAT3) dephosphorylation in human colon cancer cells[42]. These results are consistent with our findings in OSCC cells.

EP4 and CaMKK2 are known to correlate with the progression and survival rates in cancers such as vulvar cancer and breast cancer[43,44]. To the best of our knowledge, there have been no reports in oral cancer linking EP4, CaMKK2, or CALML6 with cancer progression or survival rates, indicating a need for further investigation (Supplementary Fig.16)[45].

In summary, our study reaffirms the significance of CALML6 and the EP4/CALML6/CaMKK2/AMPK signaling pathway in OSCC progression,

offering a comprehensive overview of their roles in cancer cell migration and suggesting them as potential targets for OSCC therapy. The detailed exploration of these molecules and pathways lays the groundwork for future research aimed at unraveling the complexities of OSCC and developing more effective treatments.

## Methods
### Reagents
The EP4 agonist ONO-AE1-437 was kindly provided by Ono Pharmaceutical Co., Ltd (Osaka, Japan)[9]. The EP4 agonist was used at a concentration of 1 µM in all experiments. The CaMKK2 inhibitor (STO-609) and the SOCE inhibitor (YM58483) were purchased from Cayman Chemical Company (MI, USA). N-Acetyl-L-cysteine (NAC) was purchased from Wako (Osaka, Japan).

### Cell lines
The human oral squamous cell carcinoma cell lines HSC-3 and OSC-19, which are high metastatic potential lines, were purchased from the Health Science Research Resources Bank (Japan Health Sciences Foundation, Tokyo, Japan)[9,46]. The human oral squamous cell carcinoma cell lines SAT and HSC-4 were acquired from the Japanese Collection of Research Bioresources (Osaka, Japan). The SAT cell line exhibits non-metastatic characteristics, whereas HSC-4 is characterized by a low rate of metastasis[10,24]. The human glioblastoma cell line LN229 was purchased from American Type Culture Collection (ATCC) (Virginia, USA). These cell lines were cultured in Dulbecco's modified Eagle's medium (DMEM [High Glucose], WAKO, Osaka, Japan) supplemented with L-glutamine and phenol red or in sodium pyruvate medium containing 10% fetal bovine serum (FBS), 1% penicillin–streptomycin and 1% L-glutamine. Human gingival fibroblasts (HGnF cells) and human oral keratinocytes were purchased from ScienCell Research Laboratories (CA, USA).

### Western blotting
Western blot analyses were performed as using the following method[9,46]. Briefly, cells were lysed and scraped in RIPA buffer (Thermo Scientific, IL, USA). The following primary antibodies were used for immunoblotting: anti-CaMKK2, anti-phospho-CaMKK2, anti-AMPK, and anti-phospho-AMPK, anti-MLC and anti-phospho-MLC, which were obtained from CST (Cell Signaling Technology, MA, USA); anti-GAPDH, which was obtained from Santa Cruz Biotechnology (CA, USA); anti-EP4, which was obtained from Cayman (MI, USA); anti-CALML6, which was obtained from Proteintech (IL, USA); and anti-PGC1-α, which was obtained from Abcam (Cambridge, UK). Chemiluminescence detection was performed using ECL reagent (Bio-Rad Laboratories, CA, USA) and high-sensitivity ECL reagent (Thermo Scientific, IL, USA). Signals were visualized using a LuminoGraph II imaging system (ATTO, Tokyo, Japan). The signal intensities of the bands were quantified using ATTO CS Analyzer 4 software (ATTO).

### Fluorescence imaging of intracellular $Ca^{2+}$
Measurement of the intracellular $Ca^{2+}$ concentration was performed as previously described with some modifications[7,9]. HGnF cells and HSC-3 cells were incubated with 2 µM Fura 2-AM (1-[6-amino-2-(5-carboxy-2-oxazolyl)-5-benzofuranyloxy]-2-(2-amino-5-methylphenoxy)ethane-N,N,N',N'-tetraacetic acid) and 4 µM Fluo 4-AM (1-[2-amino-5-(2,7-difluoro-6-acetoxymethoxy-3-oxo-9-xanthenyl)phenoxy]-2-(2-amino-5-

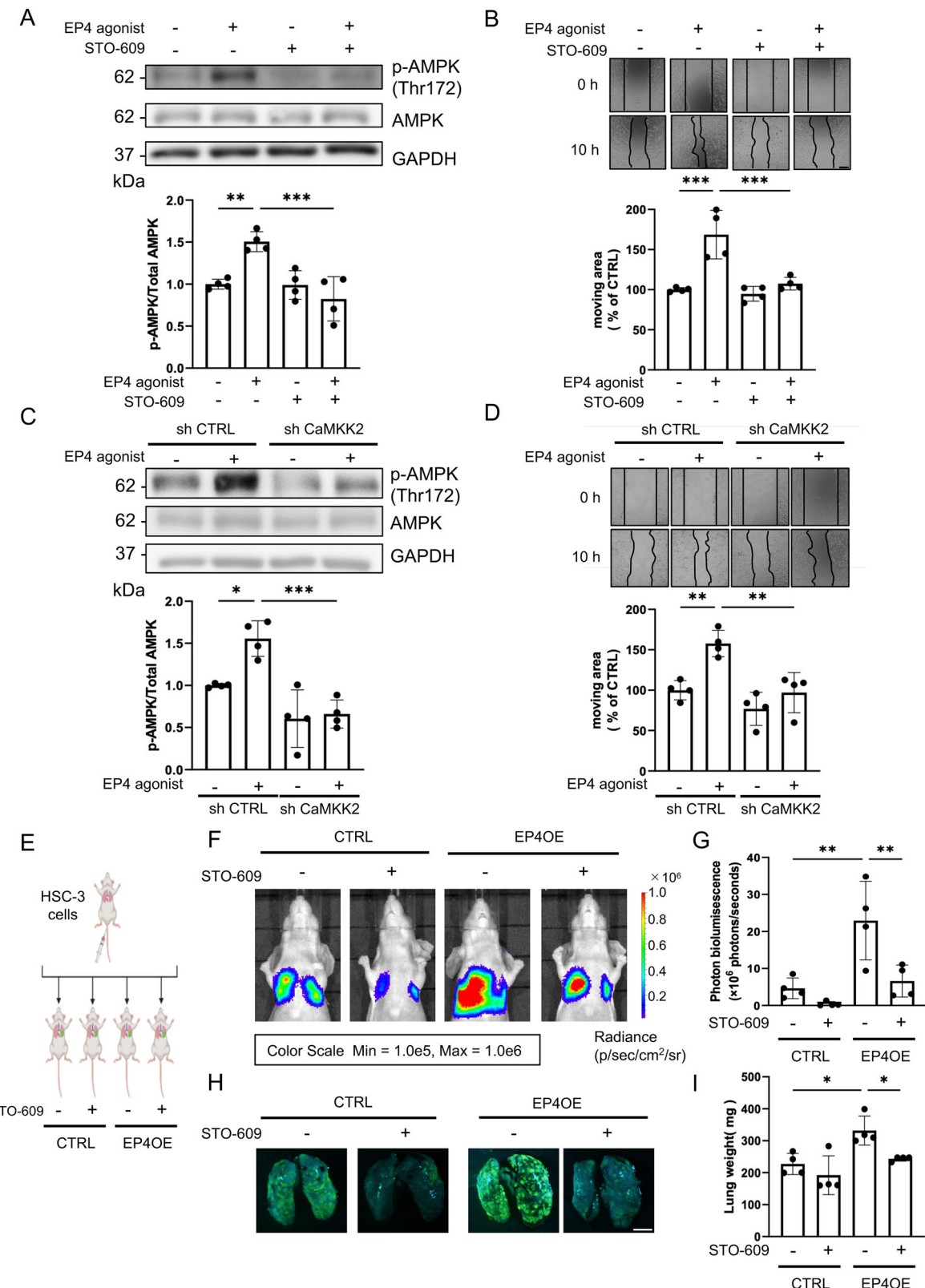

methylphenoxy)ethane-N,N,N',N'-tetraacetic acid) (Dojindo Laboratories, Kumamoto, Japan)/2-[4-(2-hydroxyethyl)-1-piperazinyl] ethanesulfonic acid (HEPES) buffer for 20 min at 37 °C. In the Fura 2-AM assay, measurements were performed using a microplate reader (Nivo, PerkinElmer, MA, USA). In the Fluo 4-AM assay, measurements were performed using a microscope (ECLIPSE Ti, Nikon Corporation, Tokyo, Japan).

**Scratch wound healing assay**

The scratch assay was performed using the following method[7,9]. Bright-field images were acquired (ECLIPSE Ti, Nikon Corporation, Tokyo, Japan) and analyzed. Cells were seeded in 24-well plates (2.0-3.0 × 10^5 cells per well) and incubated at 37 °C. At 100% confluence, the cell monolayers were scratched with sterile 1000 μl pipette tips and washed with medium to remove any

**Fig. 5 | STO-609, a CaMKK2 inhibitor, was involved in the EP4-induced phosphorylation of AMPK, cell migration and lung metastasis in vivo.**
**A** Representative images of AMPK in the presence or absence of the EP4 agonist are shown (*upper*). Densitometric analysis of western blot bands showed that STO-609 abolished the EP4-induced AMPK phosphorylation in HSC-3 cells (*lower*).
**B** Representative images of the scratch assay in the presence or absence of the EP4 agonist are shown (*upper*). The densitometric analysis of the scratch assay showed that STO-609 decreased the EP4-induced migration of HSC-3 cells (*lower*).
**C** Representative images of p-AMPK in the presence or absence of the EP4 agonist are shown (*upper*). Densitometric analysis of western blot bands showed that knockdown of CaMKK2 decreased EP4-induced AMPK phosphorylation (*lower*).
**D** Representative images of the scratch assay in the presence or absence of the EP4 agonist are shown (*upper*). Densitometric analysis of western blot bands showed that knockdown of CaMKK2 decreased cell migration (*lower*). **E** Illustrating the schema for the animal experiment. HSC-3 cells were administered through the mice tail vein. Both the CTRL and overexpression of EP4 groups underwent treatment with the STO-609, after which the presence or absence of lung metastasis was observed. The schema was created using BioRender. **F** IVIS images of lung metastasis in mice in each STO-609 (30 μg/kg body weight) treatment group on Day 21. **G** Photon bioluminescence in each group is shown on Day 21 using IVIS. **H** GFP fluorescence images illustrating lung metastasis in mice from each treatment group on day 21. **I** The graph shows the lung weight of each group on 21 days. **A–D, F–I** Data are representative of $n = 4$ independent experiments. **A–D, G, I** One-way ANOVA, Tukey's multiple comparisons test; *$p < 0.05$, **$p < 0.01$, ***$p < 0.001$. Source data and exact $p$ values are provided as a source data file.

detached cells. Images were acquired at 0 and 10 h, and the cell migration distance was calculated and analyzed.

## Real-time cell migration assay
In vitro cell migration was assessed using the xCELLigence Real-Time Cellular Analysis (RTCA) system (ACEA Biosciences, San Diego, CA, USA). After assembling the CIM-Plate 16 and allowing for a 1-hour equilibration period in an incubator, 50,000 cells in standard growth medium supplemented with 1% fetal bovine serum (FBS)—reduced from the standard 10% to minimize proliferation were added to each well. The EP4 agonist was introduced 30 minutes following cell seeding, and the Cell Index (CI) was employed for normalization purposes. Data were acquired at 15-minute intervals. Impedance changes, expressed as the Cell Index, were automatically quantified as live cells interacted with the electrodes in the E-Plates, which correlates with cell migration over time.

## Cell tracking migration assay
Bright field images were acquired using an ECLIPSE Ti microscope (Nikon Corporation, Tokyo, Japan) and subsequently analyzed. Cells were seeded in 24-well plates at a density of 10,000 cells per well and incubated at 37 °C. Images were captured at 10-minute intervals over a period of 5 hours, and the nuclei of migrating cells were marked using the manual tracking feature of NIS-Elements software (version 4.3). The speed and range of cell migration were calculated and analyzed. For assessing directionality, the X and Y coordinates of migrating cells were transferred to a polar plot.

## Invasion assay
Cells ($2.0 \times 10^5$/ml) were seeded on the top surface of a polycarbonate Transwell filter coated with Matrigel (for the Transwell invasion assay) in the top chamber of the plate provided with the QCM 24-Well Cell Invasion Assay (Cell Biolabs, Inc., CA, USA)[7,47]. Cells were suspended in a medium without serum in the top chamber, and a medium with 10% FBS was placed in the bottom chamber. The cells were incubated at 37 °C for 24 h. The noninvaded cells in the top chambers were removed with cotton swabs. The invaded cells on the lower membrane surface were fixed with 4% paraformaldehyde and stained with crystal violet. Cells in four random non-overlapping fields were counted under a microscope (BZ-X800, Keyence, Tokyo, Japan).

## Cell viability assay
HSC-3 cells were seeded in a 96-well plate at a density of $5 \times 10^3$ cells per well. The cells were cultured in DMEM for 24 h. Cell viability was measured by using Cell Counting Kit-8 (Dojindo, Kumamoto, Japan) according to the manufacturer's protocols.

## Immunocytochemistry
Immunocytochemistry was performed using the following method[7]. Filamentous actin (F-actin) staining was performed by incubation with rhodamine phalloidin (Cytoskeleton, Inc., CO, USA; cat# PHDR1). Images were acquired with a microscope (BZ-X800), and cells with one or more lamellipodia were counted manually. The extent of lamellipodia at cell edges

was measured in a minimum of 50 cells across five randomly selected fields of view. Values represent data from three independent experiments.

## In vitro retroviral transduction
Retroviral transduction of HSC-3 cells was carried out with the luciferase gene, using the following method[48]. The expression of luciferase was confirmed using the IVIS system (PerkinElmer, IL, USA).

## In vitro lentiviral transduction
The Lentiviral construct pLV[Exp]-EGFP:T2A:Bsd-EF1A > hPTGER4 was synthesized by VectorBuilder (Chicago, USA) and transfected into HSC-3 cells. After 24 h, cell culture medium was replaced, and stably transfected cells were selected using 0.15 μg/mL blasticidin. A scramble control was also purchased from VectorBuilder (Chicago, USA). The transfection efficiency of the lentivirus was assessed by RT-qPCR.

## Immunofluorescence staining
Cells were cultured to 80% confluence on 12-mm coverslips, rinsed with phosphate-buffered saline (PBS), and fixed with 2% formaldehyde for 15 minutes. The coverslips were preincubated for 30 minutes at room temperature in 0.01% Triton X-100 (PBS-Triton) to permeabilize the cells. Subsequently, the cells were incubated with the primary antibody (EP4 antibody, 1:100 dilution) overnight at 4 °C. Following this, cells were incubated with Alexa Fluor 594-conjugated goat anti-rabbit IgG (1:1000 dilution) as the secondary antibody for 1 hour at room temperature. Nuclei were stained with DAPI (1:5000 dilution) to facilitate cell nucleus detection. Fluorescence microscopy observations were made using an inverted microscope (BZ-X800, Keyence, Tokyo, Japan).

## Short hairpin RNA transduction
HSC-3 cells were subjected to transduction with CALML6 shRNA, CaMKK2 shRNA, and scramble control shRNA. Lentiviral transduction by VectorBuilder (IL, USA), Sigma-Aldrich (MO, USA), and SCBT (Santa Cruz Biotechnology, CA, USA) was performed using the following method[7,9]. The transfection efficiency of the shRNAs was evaluated by RT-qPCR. Green fluorescent protein (GFP) fluorescence was detected with a fluorescence microscope. The sequences are shown in Supplemental Table 1.

## Orthotopic xenograft tongue tumor model
HSC-3 ($4.5 \times 10^4$ cells/30 μl) cells transduced with the lentiviral scramble control shRNA or EP4 overexpression shRNA were inoculated into the left edge of the tongue of BALB/c Slc-nu/nu mice (female, 4-5 weeks old, 4 mice/group) (SLC, Shizuoka, Japan). Four weeks after inoculation of the cells, the cervical lymph nodes and the tongue were excised and fixed with formalin.

## In vivo real-time optical imaging and analysis
Real-time tumor growth was monitored by optical imaging using an IVIS-cooled CCD optical system. Mice were anesthetized using 3% isoflurane after intraperitoneal injection of 150 mg/kg body weight D-luciferin (PerkinElmer). Five minutes after the injection of D-luciferin, images were

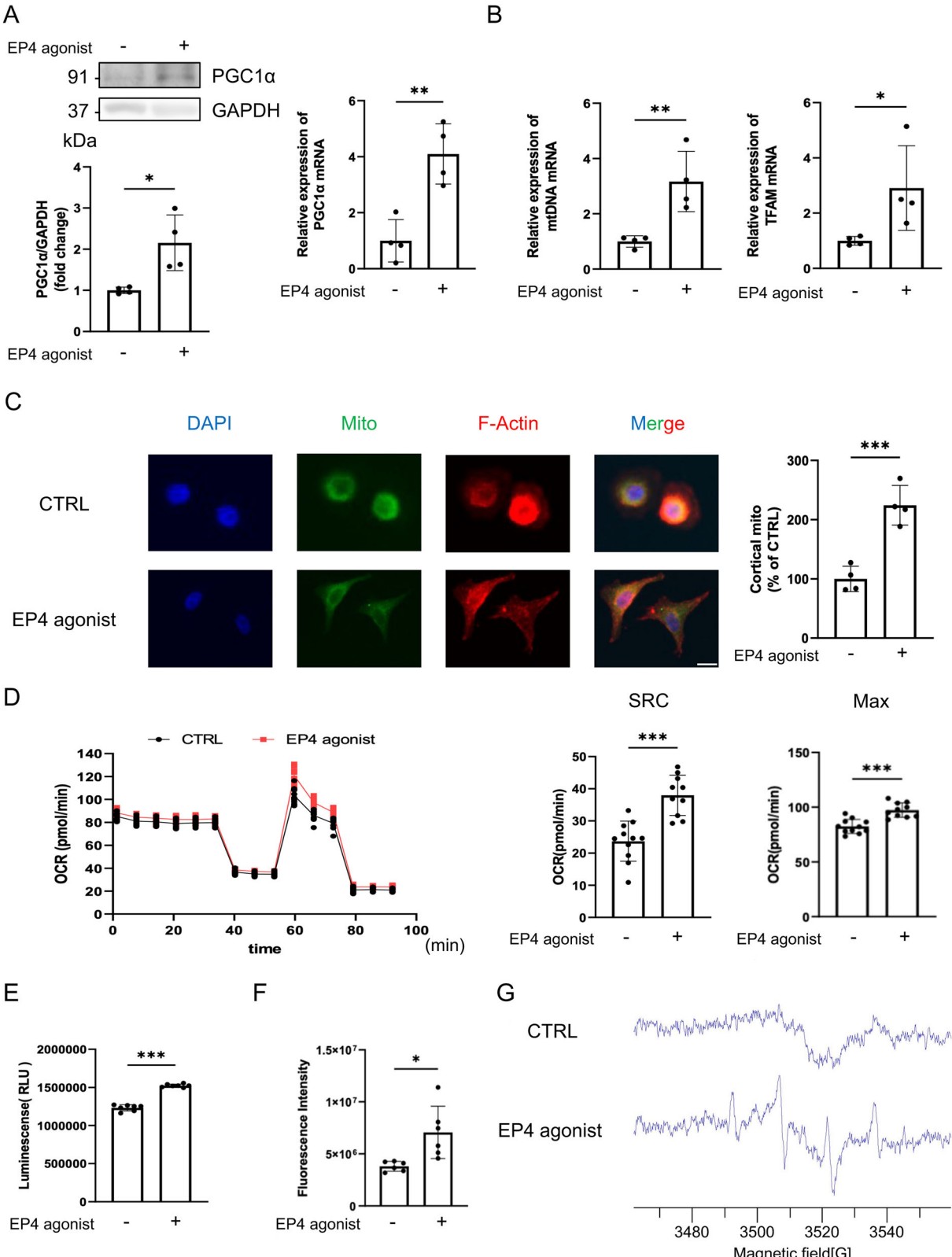

acquired for 10 to 30 sec using the analysis and acquisition software Living Image (PerkinElmer). A photographic image was acquired, onto which a pseudocolor image representing the spatial distribution of photon counts was projected. For plotting bioluminescence imaging (BLI) data, the photon flux was calculated for each mouse by using a square region of interest encompassing the head of the mouse in a supine position.

## RNA sequencing

Total RNA from HSC-3 cells was extracted using the ISOSPIN Cell & Tissue RNA Kit (Nippon Gene, Tokyo, Japan)[46]. The quality of the raw paired-end sequencing reads was assessed with FastQC. Poly(A) RNA preparation was performed using a Poly(A) mRNA Magnetic Isolation Module (New England Biolabs, MA, USA). Library preparation was performed using the

**Fig. 6 | EP4 regulated mitochondrial biogenesis and oxidative phosphorylation in HSC-3 cells. A** PGC1-α protein expression in HSC-3 cells with or without EP4 agonist treatment (*upper*). Densitometric analysis of western blot bands showed that the EP4 agonist increased the protein expression of PGC1-α (*lower*). PGC1-α mRNA transcription in HSC-3 cells with or without EP4 agonist treatment (*upper*). **B** mtDNA and TFAM mRNA transcription in HSC-3 cells with or without EP4 agonist treatment. **C** EP4 agonist treatment resulted in the translocation of mitochondria in HSC-3 cells to the edge of the cells. The left panel shows DAPI staining (*blue*) of HSC-3 cells in the presence/absence of the EP4 agonist. The second image from the left shows mitochondrial staining (*green*). The third images from left shows F-actin staining (*red*). The right panel shows triple staining with DAPI (*blue*), staining for mitochondria (*green*), and immunostaining for F-actin (*red*). The graph shows the number of lamellipodia in HSC-3 cells. More than 50 cells were counted per field of view. **D** Graphs of the data generated with the Seahorse XFe96 Extracellular Flux Analyzer in the presence or absence of the EP4 agonist are shown (*left*). The SRC (*middle*) and Maximal OCR (*right*) are shown. **E** ATP production as quantified by luminescence measurement 3 h after stimulation with the EP4 agonist. EP4 agonist treatment increased ATP production. **F** ROS production as quantified by fluorescence measurement 3 h after stimulation with the EP4 agonist. EP4 agonist treatment increased ROS production. **G** ESR analysis of ROS production 15 min after stimulation. ROS production was evaluated using an EMX-8/2.7 ESR spectrometer with/without EP4 agonist treatment. **A–G** Data are representative of $n = 4$ (**A–C, G**), $n = 6$ (**E, F**), or $n = 10\text{-}11$ (**D**) independent experiments. **A–F**, Unpaired $t$ test; $*p < 0.05$, $**p < 0.01$, $***p < 0.001$. Source data and exact $p$ values are provided as a source data file.

NEBNext® Ultra™II Directional RNA Library Prep Kit for Illumina® (New England Biolabs, MA, USA). RNA sequencing was performed using the NovaSeq 6000 system (Illumina Inc., San Diego, CA). The raw read counts were normalized by the relative log normalization (RLE) method, and differential expression analysis was conducted with DESeq2 (Version 1.24.0). Differentially expressed genes (DEGs) were identified with threshold criteria of |log2FC (fold change)| > 1 and Benjamini–Hochberg (BH) adjusted $p$-value < 0.05. Gene Ontology (GO) enrichment analysis of the DEGs was performed with GOATOOLS (Version 1.1.6). All $p$ values were corrected using the Benjamini–Hochberg method for controlling the false discovery rate.

## Quantitative real-time reverse transcription–polymerase chain reaction
Total RNA from HSC-3 cells and HGnF cells was extracted using the ISOSPIN Cell & Tissue RNA kit (Nippon Gene, Tokyo, Japan)[46]. The cDNA synthesis was performed using the PrimeScript RT Reagent Kit (TaKaRa Bio, Shiga, Japan). Reaction systems for quantitative polymerase chain reaction (qPCR) were prepared using TB Greeen Fast qPCR Mix (TaKaRa Bio). Reverse transcription–polymerase chain reaction was performed on the StepOnePlus Real-Time PCR System (Applied Biosystems, MA, USA). The 2 ΔΔCt method was used to determine relative gene expression levels, and the 18 S rRNA gene was used to normalize target gene expression levels. The primer sequences used for amplification of human genes are shown in Supplemental Table 2.

## Immunohistochemistry
Immunohistochemical staining was performed using the following method[49]. The protein expression of CALML6 was examined utilizing human heart tissue obtained from TissueArray.Com LLC, MD, USA (Catalogue No. HuFPT056). The protein expression of CALML6 was analyzed using human normal tongue tissue and a human tongue tumor tissue microarray (TissueArray.Com, MD, USA; Cat.# OR601c). Negative control samples were incubated only with the secondary antibody. Images were acquired with a microscope (BZ-X800).

## Protein-protein interaction (PPI) network construction
STRING (https://string-db.org/) is a database of known and predicted PPIs[26]. The interactions include direct (physical) and indirect (functional) associations; they originate from computational prediction, knowledge transfer between organisms, and interactions aggregated from other (primary) databases. We imported the interaction target into STRING to obtain the PPI information and visualize the network.

## Lung metastasis mouse model
HSC-3 cells transfected with the lentiviral scramble control or EP4 overexpression were harvested and injected ($2 \times 10^6$ cells/0.2 mL) into the tail veins of BALB/c Slc-nu/nu mice (female, 4–5 weeks old, 4 mice/group) (SLC)[9]. Each group of mice was administered either vehicle (0.1% dimethyl sulfoxide in phosphate-buffered saline) or STO-609 by intraperitoneal injection (30 μg/kg/body weight) twice per week for 3 weeks. Three weeks after the injection of the cells, the lungs were fixed with formalin, and the weights were measured.

## XF Cell Mito Stress Test
A total of $3 \times 10^4$ cells/well were seeded into XF Cell Culture Microplates (Agilent Technologies, CA, USA) and incubated for one day. After incubation, the cells were treated with different concentrations of the EP4 agonist for 3 h. Mitochondrial stress testing, including measurement of the mitochondrial spare respiratory capacity, was conducted according to the instructions provided by Seahorse Bioscience (Agilent)[50,51]. Briefly, cells were metabolically perturbed by the addition of three compounds in succession, and the OCR ($O_2$ consumption rate) was measured prior to and after the addition of each compound: 1.5 μM oligomycin, 0.25 μM carbonyl cyanide-4-(trifluoromethoxy) phenylhydrazone (FCCP), 0.5 μM a mixture of rotenone and 0.5 μM antimycin A. Data were analyzed using the Seahorse XFe96/XF96 Analyzer.

## Mitochondrial localization analysis
HSC-3 cells were incubated with MitoBright LT Green (Dojindo) for 30 min at 37 °C. The cells were fixed with 4.0% paraformaldehyde for 15 min, permeabilized in 0.1% Triton X-100 for 15 min, and further incubated with DAPI (Thermo Fisher) for 10 min. After three washes in PBS, coverslips were mounted with ProLong Gold (Invitrogen). For quantification of imaging data, a minimum of 50 cells per experiment were imaged using a microscope (BZ-X800). The stained area of mitochondria and the stained area of the nucleus were superimposed. Then, the area of mitochondria localized in the cytoplasm was calculated.

## ATP Assay
Intracellular ATP levels were measured by a luminescence method using the CellTiter-Glo 2.0 Luminescent Cell Viability Assay (Promega) according to the manufacturer's instructions. Briefly, $2 \times 10^4$ cells/100 μl were seeded into each well of a 96-well plate. After the addition of 100 μl of CellTiter-Glo reagent, luminescence was measured using a microplate reader (Nivo, PerkinElmer).

## Mitochondrial reactive oxygen species (ROS) measurement by fluorescence staining
Reactive oxygen species (ROS) production was measured by fluorescence staining. For fluorescence measurement, HSC-3 cells were incubated with or without the EP4 agonist. The cells were superfused with 2-(2,7-dichloro-3,6-diacetyloxy-9H-xanthen-9-yl)-benzoic acid (DCFH-DA; Cayman Chemical, MI, USA) in a serum-free medium for 30 min at 37 °C. The staining intensity was measured using a microplate reader (PerkinElmer, MA, USA). Fluorescence signals at Ex. 485 and Em. 535 nm were measured with a fluorescence microscopy system.

## ESR analysis of ROS production
ROS production was evaluated using an EMX-8/2.7 ESR spectrometer (Bruker Biospin, MA, USA) with/without EP4 agonist treatment[52]. 5,5-dimethyl-1-pyrroline-N-oxide (DMPO) solution (LABOTEC, TOKYO,

JPN) was added to the cell suspension containing HSC-3. and injected into a capillary tube. The sample was injected into a capillary tube and inserted into an ESR measurement device and measured. Testing parameters were set as follows: 20 mW microwave power, 9.85 GHz microwave frequency, 30 sec conversion time, 5 times scans, 2 G field modulation, and 100G scan range.

## Kaplan-Meier plot analysis using a publicly available dataset

OncoLnc (http://www.oncolnc.org/) was utilized to conduct a prognostic analysis for patients with head and neck squamous cell carcinoma (HNSCC) from The Cancer Genome Atlas (TCGA) dataset[46]. Patients were stratified into two groups according to the expression levels of EP4 and CaMKK2. These groups were defined as the high expression group (top 25% of expression levels) and the low expression group (bottom 25% of expression levels).

## Statistics and Reproducibility

Statistical analysis was performed using GraphPad Prism 9 software (GraphPad Software Inc., CA, USA). Statistical comparisons between two groups were performed using Student's t test. Comparisons among more than two groups were performed using one-way analysis of variance (ANOVA) followed by Tukey's test or by two-way ANOVA followed by the Bonferroni post hoc test. The criterion for statistical significance was set at $p < 0.05$.

## Reporting summary

Further information on research design is available in the Nature Portfolio Reporting Summary linked to this article.

## Data availability

The datasets generated during and/or analysed during the current study are available from the corresponding author on reasonable request.

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

## Acknowledgements
The authors are grateful to Kohei Osawa, Megumi Uchino, Fumina Suzuki, Wakana Fukae, Chihiro Hayashi, Junko Arai, Yuko Hidaka, Makiko Yamada and Mieko Niwa for their assistance with technical aspects. This study was supported in part by the Japan Society for the Promotion of Science (JSPS) KAKENHI Grant (22K06928, 22K10154 to M.U. and 22H03926 to Y.I); the Japan Agency for Medical Research and Development (AMED) (19191258, 23810577 to M.U.); the Japan Science and Technology Agency (JST) (JPMJFR205A to M.U.).

## Author contributions
S.I., M.U., K.M. and Y.I. designed the whole study and wrote the manuscript. S.I., R.N., A.N., K.N. and Y.M. conducted the pharmacological and molecular-biological studies. S.I., R.N., K.S., and M.K. conducted the animal studies. M.U., S.I. and Y.I. prepared the manuscript.

## Competing interests
The authors declare no competing interests.

## Ethics
Animal experiments were performed according to the Yokohama City University guidelines for experimental animals. All experimental protocols were approved by the Animal Care and Use Committee at Yokohama City University School of Medicine (F-A-22-044).
