## [Peer Review File · Communications Biology]

Reviewers' comments:

Reviewer #1 (Remarks to the Author):

The study by Ishikawa et al. reports that EP4, one of the four subtypes of prostaglandin E2 (PGE2) receptor regulates migration of oral squamous cell carcinoma cells via CALML6/CaMKK2/AMPK. The study employs HSC-3 and HGnF cells to report that EP4 modulates cell migration via Ca²⁺ signaling. Further, RNA sequencing and protein-protein interaction network analysis was employed to identify association between EP4, calmodulin-like protein 6 (CALML6) and calcium/calmodulin-dependent protein kinase kinase 2 (CaMKK2). Lastly, animal models were used to confirm migratory potential of EP4 overexpressing HSC-3 cells to the draining lymph nodes and lung metastasis by injecting in tail vein. Although the role of EP4 on mitochondrial function is known in white adipose tissue, its function in cancer is relatively unknown and underexplored.

In general, the language needs revision and editing throughout the manuscript.

1. Abstract:

Major comments:

The abstract is not coherent and lacks presentation of concrete aims of the study.

No mention of in vivo experiments.

Minor comments:

Line 13: The word 'established' should be replaced. The link between CALML6 is established by PPI and authors have followed up on that information.

Line 11: does not add any more information to the abstract

2. Introduction:

Major comments:

The introduction is too long and at several places too redundant and lacks coherence.

Minor comments:

Page 4, line 1: Please provide a recent reference.

Line 2-10: Needs references.

Page 5, line 10: secondary in place of second

3. Results:

Major comments:

The reviewer appreciates that the authors have provided full blots for western blotting. However, there are several positive bands for almost all the antibodies probed. The reviewer recommends having a positive control for each blot. For ex., a blot with HOK, HGnF, HSC-3, HSC-3 EP4OE, OSC-19 and LN229 probed for EP4.

Please supplement western blotting data of EP4 overexpression with immunofluorescence since total EP4 levels may not point to functional importance.

HSC-3 and OSC-19 are isolated from metastatic oral SCC. It would be interesting to see the levels of EP4 in primary OSCC cancer cell lines. The authors are recommended to use compare their findings in HSC-3

and OSC-19 with cells isolated from primary tumors.

HGnF are human gingival fibroblasts and have different gene and protein profile than epithelial cells HSC-3 that are used as model system. The reviewer recommends that experiments must be conducted by comparing HSC-3 and HOK.

Overexpression of EP4 promotes the migration of HSC-3 oral cancer cells is already shown by the authors in their previous paper: <https://www.ncbi.nlm.nih.gov/pmc/articles/PMC6942437/>. How does these results add to prior knowledge.

How many cells were analyzed to evaluate increase in number of lamellipodia by immunocytochemistry? What does n=3 in the figure legend mean?

Please provide volcano plots of RNA-seq analysis. Did the authors see any differences in gene expression in migration related and mitochondria related genes? This would be interesting as this will directly point to the role of EP4 in migration of OSCC cells.

Again, CALML6 mRNA and protein levels must be compared with HOK since these are normal keratinocytes. Comparing fibroblasts from gingiva and oral cancer keratinocytes (HSC-3, which is a highly metastatic cell line) cannot form the basis for conclusion that the authors have drawn. The reviewer suggests comparison of HSC-3 with a non-metastatic OSCC cell line that will further support the observations.

CALML6 protein levels as shown by western blotting again has several bands. The reviewer again recommends having a positive control.

The immunohistochemistry against CALML6 in mice tissue is not satisfactory. Overall, there is too much background and the antibody does not look specific. Again, a positive control would have helped.

The reviewer recommends that all scratch wound migration assays be reanalyzed based on percentage of wound closure versus all time points as it will remove the obscurity of calculations based on moving area.

Again western blots for Fig 4B are not conclusive enough to reach any deduction.

The authors established two CaMKK2 KD HSC-3 cell lines, but instead administered STO-609 to inhibit CaMKK2 in vivo. Double mutant cell line HSC-3 EP4OE CaMKK2 KD will further support the observation and lay concrete evidence.

When cells are injected into tail vein of mice, they are often 'stuck' in the lungs and are not cleared from the lungs due to their bigger size compared to mice cells. To call this phenomenon lung metastasis should be discouraged. The language in page 11, line 5-9 should be changed.

The authors provide conclusive evidence that EP4 agonist significantly increases mRNA and protein levels of mitochondrial biogenesis related genes, however the immunocytochemistry is inconclusive. The localization of mitochondria to the edge of the cells could also be due to the dysfunction of actin assembly. The reviewer recommends performing transmission electron microscopy to look at the ultrastructure of these cells and quantify the number of mitochondria.

4. Discussion:

The discussion is too long and at several places too redundant and lacks coherence. The first paragraph should briefly recognize knowledge gaps and bring together the aims and major results of the study.

Page 14, line 21: Not completely true. There are papers that suggest importance of EP4 in mitochondrial function. Couple of them are:

<https://pubmed.ncbi.nlm.nih.gov/33647796/>

<https://pubmed.ncbi.nlm.nih.gov/28533326/>

The results are overinterpreted in the sense that the authors have shown EP4/CALML6/CaMKK2/AMPK pathway in one metastatic oral cancer cell line without evaluating clinical significance. For this to be a new target for OSCC therapy, expression levels of EP4, CALML6, CaMKK2 must be evaluated in OSCC patient cohort with survival data.

Reviewer #2 (Remarks to the Author):

The authors hypothesized that the activation of a prostaglandin E2 receptor subtype (EP4) might increase the migration and invasion of Oral Squamous Cell Carcinoma (OSCC) due to an increase in mitochondria biogenesis/respiration led by increased Ca²⁺/Calmodulin signaling. The authors previously demonstrated that EP4 is overexpressed in OSCC fragments. In this study, they also showed that this increase is observed in OSCC cell lines (HSC-3) when compared to other cell lines. The authors dissected the signaling axis associated with EP4 activation and how it would affect the migration properties of tumor cells. Overall, it is a commendable work; however, there are major points that should be addressed.

Major Points:

1. Using an agonist of EP4, the authors observed an increase in Ca signaling in OSCC but not in fibroblasts. Did the authors perform a gradient with the agonist? Additionally, in Figure 1C, the authors used the wound healing scratch model for cell migration, which is highly influenced by cell proliferation. Therefore, the authors should add an internal control regarding cell proliferation to all wound healing experiments.
2. Since the authors can perform movies of cells, it would be important to conduct other assays for cell migration and analyze migration velocity, directionality, and persistence. This would mitigate the potential impact of cell proliferation and provide insights into cell signaling.
3. In Figure 1D, the authors should change "Normal Oral Cells" to "Gingival fibroblasts."
4. In Figure 2, the authors overexpressed EP4 in OSCC cells and analyzed cell migration and invasion. It is not clear why they overexpressed EP4 in OSCC, as they already showed in Figure 1A and supplementary Figure 1 that EP4 expression is increased in OSCC compared to other cell lines, including normal keratinocytes. It would be more interesting if they prove their hypothesis by increasing EP4 expression in cells with low levels, especially oral keratinocytes, stimulating them with the agonist, and observing an increase in cell migration.
5. In Figure 2C, the authors showed an increase in cell lamellipodia by analyzing fixed cells stained for actin. This result is potentially artificial since it is not clear how many cells were analyzed and what the criteria for analysis were. Additionally, it is potentially conflicting with the proposed mechanism, as Ca signaling usually leads to an increase in the activation of non-muscle type 2 myosin, increasing cell contractility. Therefore, the authors should A) perform time-lapse movies and analyze the kymographs to measure cell lamellipodia dynamics (numbers per cell, velocity of formation, stability); B) analyze the number of stress fibers per cell as well as morphological parameters, including cell polarity.
6. The authors performed RNA-seq followed by bioinformatics analysis and observed an increase in CALML6 levels and its potential relation to CaMKK2 and AMPK phosphorylation. However, it is not clear how this axis correlates with cell migration pathways. It would be important to perform a RhoGTPase

activity assay (RhoA) or at least measure myosin light chain phosphorylation levels. This would be a nice way to show the connection with cell migration signaling.

7. Regarding experiments with mitochondria, there is clearly a change in cell polarity (control cells are rounded, and EP4 cells are elongated) in Figure 6C. Therefore, the affirmation “our results demonstrated that EP4 promoted mitochondrial biogenesis.” (page 12, line 2), mainly due to changes in the localization of mitochondria, is not correct.

8. The current experiments do not support the affirmation that “Based on our results, EP4 promoted mitochondrial respiration through mitochondrial biogenesis, resulting in ROS production. In turn, ROS promoted the migration of OSCC cells.” (page 12, line 20). There is no strong evidence of mitochondrial biogenesis, and there is no mechanistic approach to show the role of ROS in cell migration (for example: authors could have used an antioxidant).

9. The discussion should be improved. There is too much focus on Calcium signaling, but it lacks insights into the main focus of the manuscript: cell migration.

10. The authors should change the title, as there is no strong evidence regarding mitochondrial biogenesis.

Point-by-point response to the reviewers' comments and a list of the incorporated change

Review Comments to the Author

Reviewer #1 (Remarks to the Author):

The study by Ishikawa et al. reports that EP4, one of the four subtypes of prostaglandin E2 (PGE2) receptor regulates migration of oral squamous cell carcinoma cells via CALML6/CaMKK2/AMPK. The study employs HSC-3 and HGnF cells to report that EP4 modulates cell migration via Ca²⁺ signaling. Further, RNA sequencing and protein-protein interaction network analysis was employed to identify association between EP4, calmodulin-like protein 6 (CALML6) and calcium/calmodulin-dependent protein kinase kinase 2 (CaMKK2). Lastly, animal models were used to confirm migratory potential of EP4 overexpressing HSC-3 cells to the draining lymph nodes and lung metastasis by injecting in tail vein.

Although the role of EP4 on mitochondrial function is known in white adipose tissue, its function in cancer is relatively unknown and underexplored.

In general, the language needs revision and editing throughout the manuscript.

Response: Thank you for your numerous important comments of reviewer #1. I have addressed and responded to each piece of advice individually as detailed below. I would appreciate it if you could kindly review these responses.

Note: We have prepared two types of manuscript and supplementary materials (with/without comment number and red text) for convenience.

1. Abstract:

Major comments:

Comment #1

The abstract is not coherent and lacks presentation of concrete aims of the study.

No mention of *in vivo* experiments.

Response: We appreciate the reviewer's constructive feedback. In response, we have revised the abstract to include a clear presentation of our study's concrete aims. Additionally, we have incorporated details about the *in vivo* experiments conducted as part of our research. This should provide a more comprehensive and coherent overview of our study in the abstract (Page 3).

Minor comments:

Comment #2

P.3 Line 13: The word 'established' should be replaced. The link between CALML6 is established by PPI and authors have followed up on that information.

Response: Thank you for your valuable suggestion. I have revised the abstract, omitting the use of 'established' as per your recommendation (Page 3, Line10-13).

Comment #3

Line 11: does not add any more information to the abstract

Response: Thank you for your valuable suggestion. In accordance with your recommendation, I have revised the abstract to omit the mentioned information (Page 3).

2. Introduction:

Major comments:

Comment #4

The introduction is too long and at several places too redundant and lacks coherence.

Response: Thank you for your valuable suggestion. Following your guidance, I have shortened and streamlined the introduction for clarity and consistency (Page 4-6).

Minor comments:

Comment #5

Page 4, line 1: Please provide a recent reference.

Response: Thank you for your valuable suggestion. I have now included a recent reference in the introduction section of our manuscript as below.

[1] Global cancer statistics 2020: GLOBOCAN estimates of incidence and mortality worldwide for 36 cancers in 185 countries

Comment #6

Line 2-10: Needs references.

Response: Thank you for your valuable suggestion. I have now included a recent reference in the introduction section of our manuscript as below.

[2] Biological biomarkers of oral cancer

[3] The emerging role of oral microbiota in oral cancer initiation, progression and stemness

[4] Immune Modulation of Head and Neck Squamous Cell Carcinoma and the Tumor Microenvironment by Conventional Therapeutics

Comment #7

Page 5, line 10: secondary in place of second

Response: Thank you for your important suggestion. We have replaced the word as you advised on Page 5, line 10, using 'secondary' in place of 'second'(Page 4, Line22).

3. Results:

Major comments:

Comment #8

The reviewer appreciates that the authors have provided full blots for western blotting. However, there are several positive bands for almost all the antibodies probed. The reviewer recommends having a positive control for each blot. For ex., a blot with HOK, HGnF, HSC-3, HSC-3 EP4OE, OSC-19 and LN229 probed for EP4.

Response: Thank you for your comments. Due to the specificity of the antibody, multiple bands have appeared. However, by comparing the band expression levels between the knockdown cells and control cells, we have confirmed that the bands presented in the results correspond to EP4 and CALML6. We have added the original blot data for each Western blot as **Supplemental Figure 1B (EP4) and 7 (CALML6)**.

Comment #9

Please supplement western blotting data of EP4 overexpression with immunofluorescence since total EP4 levels may not point to functional importance.

Response: Thank you for your comment. In this study, we used lentivirus incorporating EGFP into the same plasmid to establish EP4-overexpressing cells. This was done to assess the efficiency of lentiviral introduction and identify lymph node metastasis in vivo. Consequently, in the Western blot results, the band for EP4 overexpression appears higher than the position of normal EP4. However, bands are visible at the anticipated positions, confirming that EP4 is overexpressed at the protein level. We have added these results to **Supplemental Figure 5C**.

Comment #10

HSC-3 and OSC-19 are isolated from metastatic oral SCC. It would be interesting to see the levels of EP4 in primary OSCC cancer cell lines. The authors are recommended to use compare their findings in HSC-3 and OSC-19 with cells isolated from primary tumors.

Response: Thank you for your valuable advice. We evaluated the protein expression of EP4 in the primary tumor cell line SAT and the low metastatic cell line HSC-4 using Western blotting as **Supplemental Figure 1C**. The results indicated that EP4 expression was highest in HSC-3. These findings suggest that, as expected, EP4 expression increases with higher metastatic potential.

For further information on the characteristics and metastatic properties of each cell line, please refer to the cited literature below.

【HSC-4】

[24] Variant sublines with different metastatic potentials selected in nude mice from human oral squamous cell carcinomas

【SAT】

[25] Suppression of metastasis by tissue inhibitor of metalloproteinase-1 in a newly established human oral squamous cell carcinoma cell line

Comment #11

HGnF are human gingival fibroblasts and have different gene and protein profile than epithelial cells HSC-3 that are used as model system. The reviewer recommends that experiments must be conducted by comparing HSC-3 and HOK.

Response: Thank you for your valuable advice. Given that HOK cells proliferate slowly and are challenging to culture, we utilized the Transwell Assay, which is suitable for small quantities of HOK cells, to evaluate cell migration. Additionally, intracellular Ca²⁺ concentration measurements were conducted under a microscope using Fluo-4.

Firstly, regarding cell migration, we observed no change in HOK cell migration ability even after EP4 stimulation. Similarly, no increase in intracellular Ca²⁺ concentration following EP4 stimulation was noted. These results indicate that, similar to HGnF, in normal oral cells, EP4 expression is low, and no changes in calcium influx or cell migration ability are evident upon EP4 stimulation. We have included these results in **Supplemental Figure 4**.

Comment #12

Overexpression of EP4 promotes the migration of HSC-3 oral cancer cells is already shown by the authors in their previous paper: <https://www.ncbi.nlm.nih.gov/pmc/articles/PMC6942437/>. How does these results add to prior knowledge.

Response: Thank you for your important observation. Allow me to explain. Our previous paper demonstrated that in oral cancer cells, EP4 stimulation regulates cell migration via Ca²⁺ influx. However, we did not investigate the mechanism by which Ca²⁺ regulates migration, nor whether actual overexpression of EP4 enhances cell migration. Therefore, in this study, we examined and demonstrated that in oral cancer cell lines with EP4 overexpression, cell migration is indeed regulated by alterations in Ca²⁺-related proteins and mitochondrial polarity, both *in vitro* and *in vivo*.

Comment #13

How many cells were analyzed to evaluate increase in number of lamellipodia by immunocytochemistry? What does n=3 in the figure legend mean?

Response: Thank you for pointing this out. We measured the extent of lamellipodia at the edges in a minimum of 50 cells across five randomly selected fields of view. This count was performed three times independently. I have added the aforementioned information to **the materials and methods section** (Page 17, Line 9-11).

Comment #14

Please provide volcano plots of RNA-seq analysis. Did the authors see any differences in gene expression in migration related and mitochondria related genes? This would be interesting as this will directly point to the role of EP4 in migration of OSCC cells.

Response: Thank you for highlighting this. We present a volcano plot and will show genes associated with differentially expressed genes (DEGs) in a separate document. (**Supplementary Fig.6 and Source data**) This time, our goal was to explore genes related to calcium through RNA-seq analysis. We plan to further investigate comprehensive gene analyses related to migration and mitochondria, including taking time-course measurements, in future studies.

Comment #15

Again, CALML6 mRNA and protein levels must be compared with HOK since these are normal keratinocytes. Comparing fibroblasts from gingiva and oral cancer keratinocytes (HSC-3, which is a

highly metastatic cell line) cannot form the basis for conclusion that the authors have drawn. The reviewer suggests comparison of HSC-3 with a non-metastatic OSCC cell line that will further support the observations.

Response: Thank you for your valuable advice. We assessed CALML6 protein expression in human oral keratinocytes (HOK), the primary tumor cell line SAT, and two oral cancer cell lines with differing metastatic potentials: the low metastatic HSC-4 and the high metastatic HSC-3, using Western blotting (see **Supplementary Fig. 7**). Our results showed that CALML6 protein expression in the oral cancer cell lines was higher than in HOK. Notably, CALML6 expression was highest in the HSC-3 line. These findings suggest that, in line with our expectations, EP4 expression correlates positively with increased metastatic potential.

Comment #16

CALML6 protein levels as shown by western blotting again has several bands. The reviewer again recommends having a positive control.

Response: Thank you for your valuable comment. Regarding the Western Blot results, the presence of multiple bands is likely due to the low specificity of the antibody. However, we have tested several antibodies for CALML6, and we believe the one used is the best available currently. As we demonstrate in the supplemental data (**Supplementary Figure 9**), the bands disappear when CALML6 is knocked down, leading us to conclude that these bands are indeed representative of CALML6.

Comment #17

The immunohistochemistry against CALML6 in mice tissue is not satisfactory. Overall, there is too much background and the antibody does not look specific. Again, a positive control would have helped.

Response: As you pointed out, the background in the immunohistochemical staining for CALML6 appears low, possibly due to its relatively low expression levels, resulting in a weaker signal. However, prior to conducting this experiment, we confirmed that this antibody works as specified in the data sheet using human heart tissue (**Supplementary Figure 8**)

Comment #18

The reviewer recommends that all scratch wound migration assays be reanalyzed based on percentage of wound closure versus all time points as it will remove the obscurity of calculations based on moving area.

Response: In our recent scratch assay, we measured the wound closure rate at every hour up to 10 hours. The rationale for concluding the measurements at 10 hours is twofold. Firstly, beyond 10 hours, the wounds in the EP4 stimulation group were completely closed, making further evaluation challenging. Secondly, we aimed to assess the wound closure within a timeframe where the cell proliferation would not significantly impact the results, and before the complete closure of wounds in the EP4 stimulation group.

Comment #19

Again western blots for Fig 4B are not conclusive enough to reach any deduction.

Response: Thank you for your valuable comment. Regarding the Western Blot results, the presence of multiple bands is likely due to the low specificity of the antibody. As we demonstrate in the **Figure 5c**, the bands disappear when CaMKK2 is knocked down, leading us to conclude that these bands are indeed representative of CaMKK2.

Comment #20

The authors established two CaMKK2 KD HSC-3 cell lines, but instead administered STO-609 to inhibit CaMKK2 in vivo. Double mutant cell line HSC-3 EP4OE CaMKK2 KD will further support the observation and lay concrete evidence.

Response: Thank you for your important observation. Your suggestion to establish a double mutant cell line HSC-3 EP4OE CaMKK2 KD for further supporting our observations is indeed valuable and would likely provide more concrete evidence. We appreciate this recommendation and are keen to explore this in future research. However, due to certain technical and resource constraints in the current study, it was challenging to implement this approach at this stage. We look forward to addressing this aspect in our subsequent work to provide a more comprehensive understanding of the subject.

Comment #21

When cells are injected into tail vein of mice, they are often 'stuck' in the lungs and are not cleared from the lungs due to their bigger size compared to mice cells. To call this phenomenon lung metastasis should be discouraged. The language in page 11, line 5-9 should be changed.

Response: Thank you very much for your valuable advice. I must humbly state that the lung metastasis model used in our experiment has been widely utilized in previously published papers, which is why we also chose to use it. As you pointed out, it is not a model that accurately reflects lung metastasis. Therefore, following your advice, we have revised the term to 'experimental' lung metastasis model

(Page 10, Line 9-14).

Comment #22

The authors provide conclusive evidence that EP4 agonist significantly increases mRNA and protein levels of mitochondrial biogenesis related genes, however the immunocytochemistry is inconclusive. The localization of mitochondria to the edge of the cells could also be due to the dysfunction of actin assembly. The reviewer recommends performing transmission electron microscopy to look at the ultrastructure of these cells and quantify the number of mitochondria.

Response: Thank you for your valuable advice. Unfortunately, we do not have access to a transmission electron microscope in our lab or within our facility, which prevented us from immediately conducting the experiment. Additionally, we attempted to use the microscopes available in our lab, but faced difficulties in obtaining the necessary reagents and lenses. Therefore, we plan to consider this approach once the required materials become available in the future.

4. Discussion:

Comment #23

The discussion is too long and at several places too redundant and lacks coherence. The first paragraph should briefly recognize knowledge gaps and bring together the aims and major results of the study.

Response: Thank you for your valuable suggestion. Following your guidance, I have shortened and streamlined the discussion for clarity and consistency (Page 12-14).

Comment #24

Page 14, line 21: Not completely true. There are papers that suggest importance of EP4 in mitochondrial function. Couple of them are:

<https://pubmed.ncbi.nlm.nih.gov/33647796/>

<https://pubmed.ncbi.nlm.nih.gov/28533326/>

Response: Thank you for your important observation. Additionally, I appreciate your providing two significant references. In light of these, I have included these references and revised the text accordingly.

[42] Prostaglandin E receptor subtype 4 protects against diabetic cardiomyopathy by modulating cardiac fatty acid metabolism via FOXO1/CD36 signalling

[43] Prostaglandin E receptor subtype 4 regulates lipid droplet size and mitochondrial activity in murine subcutaneous white adipose tissue

Comment #25

The results are overinterpreted in the sense that the authors have shown EP4/CALML6/CaMKK2/AMPK pathway in one metastatic oral cancer cell line without evaluating clinical significance. For this to be a new target for OSCC therapy, expression levels of EP4, CALML6, CaMKK2 must be evaluated in OSCC patient cohort with survival data.

Response: Thank you for your important observation. When compiling data on the survival rates of EP4 and CaMKK2 in Head and Neck Squamous Cell Carcinoma from the TCGA database, the results were as follows. However, no significant difference in survival rates was observed (**Supplementary Fig.16**). Unfortunately, we could not obtain data for CALML. Additionally, as this dataset includes all head and neck cancer patients, it may not be entirely appropriate for analyzing specifically oral cancer patients. In the future, we plan to conduct studies using clinical patient samples from our affiliated hospital to further investigate this matter.

Reviewer #2 (Remarks to the Author):

The authors hypothesized that the activation of a prostaglandin E2 receptor subtype (EP4) might increase the migration and invasion of Oral Squamous Cell Carcinoma (OSCC) due to an increase in mitochondria biogenesis/respiration led by increased Ca²⁺/Calmodulin signaling. The authors previously demonstrated that EP4 is overexpressed in OSCC fragments. In this study, they also showed that this increase is observed in OSCC cell lines (HSC-3) when compared to other cell lines. The authors dissected the signaling axis associated with EP4 activation and how it would affect the migration properties of tumor cells. Overall, it is a commendable work; however, there are major points that should be addressed.

Response: Thank you for your numerous important comments of reviewer #2. I have addressed and responded to each piece of advice individually as detailed below. I would appreciate it if you could kindly review these responses.

Major Points:

Comment #26

1. Using an agonist of EP4, the authors observed an increase in Ca signaling in OSCC but not in fibroblasts. Did the authors perform a gradient with the agonist?
2. Additionally, in Figure 1C, the authors used the wound healing scratch model for cell migration, which is highly influenced by cell proliferation. Therefore, the authors should add an internal control regarding cell proliferation to all wound healing experiments.

Response 1: We thank this reviewer for this helpful suggestion. Before conducting the main experiment, we evaluated intracellular Ca^{2+} elevation using the Fura-2 assay at concentrations of 0.1, 1, and 10 μM . It was confirmed that 1 μM resulted in the most significant increase in intracellular calcium levels. Therefore, this concentration was chosen for use in our main experiments (**Supplementary Fig. 2A**).

Response 2: In cancer cells, long-term observation can lead to wound closure due to cell proliferation. Therefore, in this experiment, we set the observation period to 10 hours, a duration without significant proliferation effects. Additionally, we assessed cell proliferation prior to this experiment using the XTT assay. Consequently, it appears that the experimental conditions, including EP4 stimulation, did not affect cell proliferation during the observation period (**Supplementary Fig. 3**).

Comment #27

Since the authors can perform movies of cells, it would be important to conduct other assays for cell migration and analyze migration velocity, directionality, and persistence. This would mitigate the potential impact of cell proliferation and provide insights into cell signaling.

Response: Thank you for your insightful comments. We share your interest in the suggested approach. Unfortunately, our current laboratory setup posed challenges in acquiring the necessary lenses and analysis software for the proposed microscopic observations. Consequently, we are unable to conduct these experiments at present. We plan to revisit this method in the future once we have access to the required resources.

Comment #28

In Figure 1D, the authors should change "Normal Oral Cells" to "Gingival fibroblasts."

Response: Thank you for your feedback. As per your instructions, I have changed the notation in the **Figure 1D**.

Comment #29

In Figure 2, the authors overexpressed EP4 in OSCC cells and analyzed cell migration and invasion. It is not clear why they overexpressed EP4 in OSCC, as they already showed in Figure 1A and supplementary Figure 1 that EP4 expression is increased in OSCC compared to other cell lines, including normal keratinocytes. It would be more interesting if they prove their hypothesis by increasing EP4 expression in cells with low levels, especially oral keratinocytes, stimulating them with

the agonist, and observing an increase in cell migration.

Response: Thank you for your valuable comment. Your suggestion to investigate the effects of increased EP4 expression in cells with low levels, particularly oral keratinocytes, is indeed compelling and represents a meaningful direction for future research. While we recognize the significance of this approach, it was challenging to include in our current study due to practical constraints, including the time-consuming process of obtaining necessary approvals for genetic modification. We are very interested in exploring this in our future work to further clarify the role of EP4 in cell migration.

Comment #30

In Figure 2C, the authors showed an increase in cell lamellipodia by analyzing fixed cells stained for actin. This result is potentially artificial since it is not clear how many cells were analyzed and what the criteria for analysis were. Additionally, it is potentially conflicting with the proposed mechanism, as Ca signaling usually leads to an increase in the activation of non-muscle type 2 myosin, increasing cell contractility. Therefore, the authors should A) perform time-lapse movies and analyze the kymographs to measure cell lamellipodia dynamics (numbers per cell, velocity of formation, stability); B) analyze the number of stress fibers per cell as well as morphological parameters, including cell polarity.

Response: Thank you for your important comment. Unfortunately, we encountered difficulties in obtaining the necessary reagents and lenses to conduct the experiment using the microscopes available in our department. Therefore, we plan to consider this approach once the required materials become available in the future.

Comment #31

The authors performed RNA-seq followed by bioinformatics analysis and observed an increase in CALML6 levels and its potential relation to CaMKK2 and AMPK phosphorylation. However, it is not clear how this axis correlates with cell migration pathways. It would be important to perform a RhoGTPase activity assay (RhoA) or at least measure myosin light chain phosphorylation levels. This would be a nice way to show the connection with cell migration signaling.

Response: Thank you for your valuable comment. We have confirmed through Western blotting that phosphorylation of myosin light chain is enhanced with EP4 stimulation. In the future, we also intend to examine Rho GTPase activity assay. We have added the results of the Western blot to the **Supplemental Figure 10**.

Comment #32

Regarding experiments with mitochondria, there is clearly a change in cell polarity (control cells are rounded, and EP4 cells are elongated) in Figure 6C. Therefore, the affirmation “our results demonstrated that EP4 promoted mitochondrial biogenesis.” (page 12, line 2), mainly due to changes in the localization of mitochondria, is not correct.

Response: Thank you for your feedback. Following your advice, I have revised the expression in the text (Page 11, Line 8-9).

Comment #33

The current experiments do not support the affirmation that “Based on our results, EP4 promoted mitochondrial respiration through mitochondrial biogenesis, resulting in ROS production. In turn, ROS promoted the migration of OSCC cells.” (page 12, line 20). There is no strong evidence of mitochondria biogenesis, and there is no mechanistic approach to show the role of ROS in cell migration (for example: authors could have used an antioxidant).

Response: Thank you for your important advice. As suggested, we measured cell migration under EP4 stimulation using the antioxidant N-acetylcysteine (NAC). The results showed that NAC inhibited the enhanced migration ability induced by EP4(**Supplementary Fig. 15**). This suggests that ROS plays a role in cell migration.

Comment #34

The discussion should be improved. There is too much focus on Calcium signaling, but it lacks insights into the main focus of the manuscript: cell migration.

Response: Thank you for your important feedback. Following your advice, I have revised the discussion section of the manuscript, shifting the focus from Ca²⁺-centric descriptions to descriptions of locomotor abilities.

Comment #35

The authors should change the title, as there is no strong evidence regarding mitochondrial biogenesis.

Response: Thank you for your guidance. Following your advice, I have revised the title.

REVIEWERS' COMMENTS:

Reviewer #1 (Remarks to the Author):

Overall the manuscript has improved immensely and the authors have answered most of the queries raised by this reviewer. Although the manuscript language has improved, it still needs some language revision. Please see my comments below.

Abstract

No major comments

Minor comments:

Line 7-8: replace 'have been' with 'is'

Line 12- add 'in OSCC' after undefined

Line 16: replace 'raised' with 'increased'

Introduction

No major comments

Minor comments:

Line 4-5, page 4: Head and Neck cancer is eighth most common not oral cancer. This line should be modified.

Line 3, page 5: replace 'suggests' with 'imparts'

Line 10, page 5: OSCC is already defined before

Line 11, page 5: 'as' should be 'a'

Line 23-25, page 5: please rephrase

Results

Major comment

Please supplement western blotting data of EP4 overexpression with immunofluorescence (IF) since total EP4 levels may not point to functional importance. Since the cells have EGFP, provide IF images to show EP4 localization.

Minor comments:

Line 9, page 6: replace 'noted' with 'reported'

Line 9, page 18: replace 'rise' with 'increase'

Line 8, page 7: please rephrase this line

Line 16-17, page 8: please rephrase this

Line 12-13, page 11: very important piece of result and should be discussed further. Please see this review: <https://www.ncbi.nlm.nih.gov/pmc/articles/PMC8848284/>

Discussion

Major comments

The results are overinterpreted in the sense that the authors have shown EP4/CALML6/CaMKK2/AMPK pathway in one metastatic oral cancer cell line without evaluating clinical significance. For this to be a new target for OSCC therapy, expression levels of EP4, CALML6, CaMKK2 must be evaluated in OSCC patient cohort with survival data.

Response: Thank you for your important observation. When compiling data on the survival rates of EP4 and CaMKK2 in Head and Neck Squamous Cell Carcinoma from the TCGA database, the results were as follows. However, no significant difference in survival rates was observed (Supplementary Fig.16). Unfortunately, we could not obtain data for CALML. Additionally, as this dataset includes all head and neck cancer patients, it may not be entirely appropriate for analyzing specifically oral cancer patients. In the future, we plan to conduct studies using clinical patient samples from our affiliated

hospital to further investigate this matter.
Please include this in the discussion section.

No minor comments.

Reviewer #2 (Remarks to the Author):

The authors improved the manuscript, but comments No. 27 and No. 30 (from the rebuttal letter) were not responded to satisfactorily. Authors answered: "Unfortunately, our current laboratory setup posed challenges in acquiring the necessary lenses and analysis software for the proposed microscopic observations", however they were able to perform time-lapse movies and the experiment could be performed with the same setup. For analysis of the time lapse, they could use free software widely used in the scientific community, such as ImageJ.

REVIEWERS' COMMENTS:

Reviewer #1 (Remarks to the Author):

Overall the manuscript has improved immensely and the authors have answered most of the queries raised by this reviewer. Although the manuscript language has improved, it still needs some language revision. Please see my comments below.

Thank you for your numerous important comments of reviewer #1. I have addressed and responded to each piece of advice individually as detailed below. I would appreciate it if you could kindly review these responses.

Abstract

Minor comments:

Comment 1.

Line 7-8: replace 'have been' with 'is'

Response: Thank you for pointing that out. I have made the corrections as instructed.

Comment 2.

Line 12- add 'in OSCC' after undefined

Response: Thank you for pointing that out. I have made the corrections as instructed.

Comment 3.

Line 16: replace 'raised' with 'increased'

Response: Thank you for pointing that out. I have made the corrections as instructed.

Introduction

Minor comments:

Comment 4.

Line 4-5, page 4: Head and Neck cancer is eighth most common not oral cancer. This line should be modified.

Response: Thank you for your important suggestion. We have replaced the word as you advised on Page 4, line 2-3.

Comment 5.

Line 3, page 5: replace 'suggests' with 'imparts'

Response: Thank you for pointing that out. I have made the corrections as instructed.

Comment 6.

Line 10, page 5: OSCC is already defined before

Response: We deleted the definition.

Comment 7.

Line 11, page 5: 'as' should be 'a'

Response: Thank you for pointing that out. I have made the corrections as instructed.

Comment 8.

Line 23-25, page 5: please rephrase

Response: Thank you for pointing that out. I have rephrased it as you indicated (Page5,

Line20-21). I have removed the parts that were too much.

Results

Major comment

Comment 9.

Please supplement western blotting data of EP4 overexpression with immunofluorescence (IF) since total EP4 levels may not point to functional importance. Since the cells have EGFP, provide IF images to show EP4 localization.

Response: Thank you for the valuable advice. We performed immunofluorescence using HSC-3 cells that were mentioned in EP4 (**Supplementary Fig. 5C**). In the case of EP4 overexpression, there were no obvious changes in localization to the cell periphery or increases in fluorescence intensity. Therefore, we were not able to identify clear functional analyses in this experiment, but we would like to continue investigating.

Minor comments:

Comment 10.

Line 9, page 6: replace 'noted' with 'reported'

Response: Thank you for pointing that out. I have made the corrections as instructed.

Comment 11.

Line 9, page 18: replace 'rise' with 'increase'

Response: Thank you for pointing that out. I have made the corrections as instructed.

Comment 12.

Line 8, page 7: please rephrase this line

Response: Thank you for pointing that out. I have removed the parts that were too much as you indicated (Page7, Line2-3).

Comment 13.

Line 16-17, page 8: please rephrase this

Response: Thank you for pointing that out. I have removed in part and rephrased it as you indicated (Page8, Line8-9).

Comment 14.

Line 12-13, page 11: very important piece of result and should be discussed further. Please see this review: <https://www.ncbi.nlm.nih.gov/pmc/articles/PMC8848284/>

Response: Thank you for introducing the valuable paper. I have cited this paper and added a sentence to the discussion section (Page13, Line8-12).

Discussion

Major comments

Comment 15.

The results are overinterpreted in the sense that the authors have shown EP4/CALML6/CaMKK2/AMPK pathway in one metastatic oral cancer cell line without evaluating clinical significance. For this to be a new target for OSCC therapy, expression levels of EP4, CALML6, CaMKK2 must be evaluated in OSCC patient cohort with survival data.

Response: Thank you for your important observation. When compiling data on the survival rates of EP4 and CaMKK2 in Head and Neck Squamous Cell Carcinoma from the TCGA database, the results were as follows. However, no significant difference in survival rates was observed (Supplementary Fig.16). Unfortunately, we could not obtain data for CALML. Additionally, as this dataset includes all head and neck cancer patients, it may not be entirely appropriate for analyzing specifically oral cancer patients. In the future, we plan to conduct studies using clinical patient samples from our affiliated hospital to further investigate this matter.

Please include this in the discussion section.

Response: Thank you for your valuable suggestion. I have revised the discussion (Page14, Line8-11).

Reviewer #2 (Remarks to the Author):

Comment 16.

The authors improved the manuscript, but comments No. 27 and No. 30 (from the rebuttal letter) were not responded to satisfactorily. Authors answered: "Unfortunately, our current laboratory setup posed challenges in acquiring the necessary lenses and analysis software for the proposed microscopic observations", however they were able to perform time-lapse movies and the experiment could be performed with the same setup. For analysis of the time lapse, they could use free software widely used in the scientific community, such as ImageJ.

Response: Thank you for your important comments of reviewer #2. I have addressed and responded to your advice as detailed below. I would appreciate it if you could kindly review these responses.

We utilize two types of microscopes depending on the experimental needs. One is used for capturing time-lapse images in bright field, such as for scratch assays. The other is suited for observing cell morphology after fixing cells and performing fluorescent staining. Reviewer 2 suggested capturing time-lapse movies and analyzing kymographs to measure the dynamics of lamellipodia (number per cell, formation rate, stability), the number of stress fibers per cell, and morphological parameters including cell polarity. To analyze kymographs and assess the number of stress fibers, equipment capable of capturing time-lapse images with a stage heater and CO₂ supply, as well as devices for evaluating cell morphology at high magnification, are considered necessary. Unfortunately, our microscopes lack these facilities, making such observations difficult.

Instead, we have considered two assays possible in our facility to study the speed, directionality, and persistence of cell migration. First, we used the xCELLigence system to check the speed and persistence of cell migration. The xCELLigence system utilizes the electrical impedance caused by cells to observe the number and speed of migrating cells in real time. EP4 stimulation increased cell migration and speed compared to the control (Supplementary Fig. 3A). We were also able to confirm the persistence of migration, which

was not possible with the scratch assay. Although long-term observations raise concerns about cell proliferation, we confirmed in advance that there was no significant effect of cell proliferation within the observation time (24h) (**Supplementary Fig. 3E**).

Next, we tracked individual cells to study the speed and directionality of cell migration. This assay also confirmed an increase in individual cell migration capabilities with EP4 stimulation (**Supplementary Fig. 3B, C**).

Although these investigations may not fully satisfy your advice, we take your important points seriously and intend to consider them in future studies.